

# Particle size traces modern Saharan dust transport and deposition across the equatorial North Atlantic

Michèlle van der Does[1], Laura F. Korte[1], Chris I. Munday[1], Geert-Jan A. Brummer[1,2], Jan-Berend W. Stuut[1,3]

5   [1] NIOZ – Royal Netherlands Institute for Sea Research, Department of Ocean Systems, and Utrecht University, Texel, The Netherlands
[2] Faculty of Earth and Life Sciences, Vrije Universiteit Amsterdam, The Netherlands
[3] MARUM – Center for Marine Environmental Sciences, University of Bremen, Germany

*Correspondence to: Michèlle van der Does (michelle.van.der.does@nioz.nl)*





**Abstract.** Mineral dust has a large impact on regional and global climate, depending on its particle size. Especially in the Atlantic Ocean downwind of the Sahara, the largest dust source on earth, the effects can be substantial but are poorly understood. This study focuses on seasonal and spatial variations in particle size of Saharan dust deposition across the Atlantic Ocean, using an array of submarine sediment traps moored along a transect at 12˚ N. We show that the particle size

decreases downwind with increased distance from the Saharan source, due to higher gravitational settling velocities of coarse particles in the atmosphere. Modal grain sizes vary between 4 and 33 μm throughout the different seasons and at five locations along the transect. This is much coarser than previously suggested and incorporated into climate models. In addition, seasonal changes are prominent, with coarser dust in summer, and finer dust in winter and spring. Such seasonal changes are caused by transport at higher altitudes and at greater wind velocities during summer than in winter. Also the

latitudinal migration of the dust cloud, associated with the Intertropical Convergence Zone, causes seasonal differences in deposition as the summer dust cloud is located more to the north, and more directly above the sampled transect. Furthermore, increased precipitation and more frequent dust storms in summer coincide with coarser dust deposition. Our findings contribute to understanding Saharan dust transport and deposition relevant for the interpretation of sedimentary records for climate reconstructions, as well as for global and regional models for improved prediction of future climate.

**Keywords** Mineral dust; Atlantic Ocean; grain size; Saharan dust transport; seasonality

## 1 Introduction

Millions of tons of mineral dust are transported from the African continent towards the Atlantic Ocean every year, with several direct and indirect effects on global climate. CALIPSO lidar measurements between 2007 and 2013 show that

annually 182 Tg of African dust leaves the African continent towards the Atlantic Ocean, 132 Tg reaches 35˚ W, and 43 Tg reaches as far west as 75˚ W (Yu et al., 2015). Approximately 140 Tg is deposited in the Atlantic Ocean between 15 and 75˚ W and 10˚ S and 30˚ N. Atmospheric mineral dust affects the atmosphere's radiation budget by scattering and absorbing incoming and reflected solar radiation, and changes cloud properties by acting as cloud condensation nuclei (Goudie and Middleton, 2001;Highwood and Ryder, 2014;Shao et al., 2011;Wilcox et al., 2010). Climatic effects are largely determined

by particle characteristics including particle size, particle shape, chemical- and mineralogical composition, and by cloud cover and the albedo of the underlying surface (Claquin et al., 2003;Goudie and Middleton, 2001, 2006;Highwood and Ryder, 2014;Otto et al., 2007;Shao et al., 2011;Sokolik and Toon, 1999). Large particles in the lower atmosphere may have a warming effect on earth's climate by absorbing reflected (long-wave) radiation (Mahowald et al., 2014;Otto et al., 2007). By contrast, small particles in the higher atmosphere may have a cooling effect, by reflecting incoming solar (short-wave)

radiation (Claquin et al., 2003;Mahowald et al., 2014). Moreover, dust deposition enhances ocean carbon cycling by delivering nutrients that stimulate phytoplankton growth (Martin and Fitzwater, 1988;Shao et al., 2011). In turn, this not only leads to increased export fluxes but also by faster transport of organic carbon to the deep ocean, as dust particles act as





mineral ballast, depending on particle size, shape and mineral density (Armstrong et al., 2002;Bressac et al., 2014;Fischer et al., 2007;Fischer and Karakas, 2009;Klaas and Archer, 2002). Both have the potential to reduce atmospheric $pCO_2$ levels (Klaas and Archer, 2002).

The distance over which mineral dust is transported depends on the transporting winds and particle characteristics including size, shape and density, which determine settling velocities. Thus, rounded quartz and feldspar particles have a greater settling velocity than platy clay minerals, and are therefore deposited closer to the source (Glaccum and Prospero, 1980;Goudie and Middleton, 2006;Mahowald et al., 2014;Stuut et al., 2005). Saharan dust is transported with the trade winds year-round, from the northwestern Sahara to the eastern Atlantic Ocean. During winter, the Harmattan trade winds
prevail, transporting dust from the central Sahara (Glaccum and Prospero, 1980;Stuut et al., 2005) at altitudes between 0 and 3 km (Tsamalis et al., 2013). In summer, when the larger land-sea temperature contrast results in large convective cells over the African continent, dust is emitted from the Sahara and Sahel. During transport towards the Atlantic Ocean, cool marine air blows in the opposite direction and lifts the warm, dusty air high up in the atmosphere. This Saharan air layer (SAL) is confined between two inversion layers, at 1 and 5 km height (Carlson and Prospero, 1972;Kanitz et al., 2014;Prospero and
Carlson, 1972;Tsamalis et al., 2013). Due to the latitudinal movement of the ITCZ (Intertropical Convergence Zone), the dust cloud over the Atlantic Ocean also migrates seasonally (Nicholson, 2000), shifting northward (10–20°N) in summer and southward (0–10°N) in winter (Adams et al., 2012;Holz et al., 2004;Moulin et al., 1997;Yu et al., 2015).

The particle size of entrained and transported mineral dust depends on source conditions including surface roughness, wind
velocity and erosion threshold, and soil characteristics including particle size, -shape, -density and soil moisture (d'Almeida and Schütz, 1983;Marticorena, 2014). After entrainment, the particle-size distributions are further modified by size-selective processes during transport and deposition (Grini and Zender, 2004). Owing to gravitational settling, dust particle size decreases with increasing distance from the source (Holz et al., 2004;Mahowald et al., 2014;Sarnthein et al., 1981;Schütz, 1980) and generally do not exceed 20 µm when transported over long distances (Gillette, 1979;Tsoar and Pye, 1987). On the
Cape Verde islands close to the Saharan source, Glaccum and Prospero (1980) found individual quartz and mica particles of up to 90 and 350 µm, respectively. However, various studies reported giant (> 62.5 µm) mineral dust particles also at much greater distances (> 10,000 km) from their source (Betzer et al., 1988;Goudie and Middleton, 2006;Mahowald et al., 2014;Middleton et al., 2001). Climate models usually do not account for such coarse particles, and generally overestimate the fine fraction (Grini and Zender, 2004;Kok, 2011). This not only results in an underestimation of the dust flux to the
oceans and in turn the fertilizing effect of the transported nutrients, it also produces errors in the sign and magnitude of radiative forcing by dust and the formation of cloud condensation nuclei. This affects weather forecasts and climate predictions, especially in dusty regions (Kok, 2011).



Due to their vastness, dust over the oceans has remained poorly studied, although specific information is required for predicting future climate and past climate reconstructions (IPCC, 2013). For the present study, we focused on a transect across the Atlantic Ocean, located directly underneath the Saharan dust cloud at 12˚ N (Yu et al., 2015). We used time-series submarine sediment traps moored at five locations along this transect, sampling synchronously at a resolution of 16 days,

Here we present the first-year results on seasonal variability over the full particle-size range, analyzing source-to-sink variation of particle size in relation to large-scale atmospheric processes. Atmospheric Saharan dust has been collected at daily resolution at Barbados for more than 50 years (Prospero and Carlson, 1970;Prospero and Nees, 1977;Prospero et al., 1981;Prospero and Nees, 1986;Prospero and Lamb, 2003). Although the longest dust record sampled to date, it is at a single and distal location relative to the Saharan source. Croot et al. (2004) sampled Saharan dust < 1 µm in fall 2002 from the

atmosphere along a transect across the Atlantic Ocean, while Stuut et al. (2005) also considered larger particles by shipboard sampling in winter 1998. Also Skonieczny et al. (2013) observed temporal changes in dust outbreaks and particle characteristics like grain size and chemistry, at a single proximal location on the western African coast. They found higher fluxes during winter, as opposed to coarser particles during summer, and attribute this to the seasonally different transporting dust layers. Similar higher fluxes of coarser-grained lithogenic particles in summer were observed by Ratmeyer et al.

(1999a;1999b), using a submarine sediment trap moored at a very proximal location just off NW Africa. Friese et al. (submitted) relate seasonal changes of dust particle size in sediment traps to regional meteorological variability such as precipitation, trade-wind speed and dust-storm events. In deep-sea sediments deposited offshore northwest Africa, Holz et al. (2004;2007), Mulitza et al. (2008) and Zühlsdorff et al. (2007) found links between dust deposition and variability in transport mechanisms, and more dust deposition in dry glacial periods than in humid interglacials, throughout the late

Quaternary.

## 2 Material and methods

Five moorings were deployed in October 2012 (Stuut et al., 2012), of which four were moored along a transect at 12˚ N across the equatorial North Atlantic Ocean, and a fifth at 13˚ N (Fig. 1A). Each mooring is equipped with two sediment traps, at depths of 1200 and 3500 meters below sea level (BSL), or "upper" and "lower", respectively (Fig. 1B, Table 1). The

sediment traps are model PPS 5/2 from Technicap that consist of a conical funnel (36˚) with a catchment area of 1 m$^2$ and an 8mm hexagonal baffle on top to maximize particle collection (U.S. GOFS, 1989) and prevent large swimmers from entering the sediment trap. Underneath the funnel, a rotating carrousel with 24 sampling cups collects discrete samples of the settling particle flux. All sediment traps operated synchronously over pre-programmed intervals of 16 days. Tilt-meters showed that the sediment traps remained nearly upright for the entire sampling period. This paper presents the results of successful

sampling by seven sediment traps on the five moorings from 19 October 2012 to 7 November 2013 (Stuut et al., 2013). These include three of the upper (1200 m) sediment traps located at mooring stations M1, M2 and M4, and four lower (3500





m) sediment traps at stations M2, M3, M4 and M5 (Fig. 1, Table 1). In addition, seafloor sediments were collected by a Multicorer at all five mooring stations, using the top centimeter for comparison with the sediment-trap samples.

Prior to the deployment of each sediment trap the sampling cups were filled with seawater collected at the deployment site depths, to which a biocide (HgCl$_2$; end-concentration 1.3 g L$^{-1}$) and a pH-buffer (borax; Na$_2$B$_4$O$_7$·10H$_2$O; end concentration

1.3 g L$^{-1}$, pH ≈ 8.5) were added, to a density slightly higher than the ambient seawater. In the laboratory each sample was sieved through a 1mm mesh to remove mostly zooplankton swimmers, then wet-split in five aliquots using a WSD10 Rotor splitter (McLane Laboratories, USA). The average weight difference between replicate aliquots of each sample is 2.4% (SD = 2.2), with 87% of all samples having a weight difference of < 5% between splits. The highest deviation was found to be 12%. For grain-size analysis, one of these aliquots was split into another 5 subsamples (1/25 of the original sample), that

were washed and centrifuged repeatedly at approximately 1800 x g with Milli-Q water to remove the HgCl$_2$, borax, and sea-salts.

Biogenic constituents were removed in three steps to isolate the insoluble or lithogenic dust fraction from all samples prior to grain-size analysis, following the procedure described by McGregor et al. (2009). Shortly, organic matter was oxidized using

H$_2$O$_2$, followed by dissolving the biogenic carbonates using HCl, and removing biogenic silica by adding NaOH. Immediately prior to the grain-size measurements sodium pyrophosphate (Na$_4$P$_2$O$_7$·10H$_2$O) was added to ensure complete disaggregation of the particles. The particle-size distributions were measured with a Coulter Laser Diffraction Particle Sizer (LS13 320) with a Micro Liquid Module (MLM) for small-volume samples, and a magnetic stirrer was used to homogenize the sample during analysis. This resulted in particle-size distributions consisting of 92 logarithmic size classes ranging from

0.375 to 2000 µm. Grain-size statistics were calculated geometrically using the graphical method of Folk and Ward (1957) using GRADISTAT (Blott and Pye, 2001).

To determine seasonal changes in dust deposition along the trans-Atlantic transect, the sediment-trap samples are grouped per season. The seasons are defined as follows: (boreal) fall includes September, October and November (SON) of 2012 and

2013, winter includes December, January and February (DJF) of 2012/2013, spring includes March, April and May (MAM) of 2013, and summer includes June, July and August (JJA) of 2013. The dates of the samples are referred to as the mid-date of each 16-day sampling period.

In order to determine the provenance of dust carrying air layers, four-day backward trajectories of air parcels were calculated

with the Hybrid Single Particle Lagrangian Integrated Trajectory (HYSPLIT) model (Draxler and Rolph, 2015), using the GDAS (0.5 degree) meteorological dataset (http://ready.arl.gov/HYSPLIT.php/). The heights of these air layers were chosen in accordance with typical winter and summer dust-carrying air layers (see below).



## 3 Results

### 3.1 Spatial trends in grain size

Modal grain sizes of the sediment traps and seafloor sediments show a pronounced downwind fining (Fig. 2). Coarsest Saharan dust was found in the easternmost trap (M1), rapidly fining westward towards M5. Also the seafloor sediments

show the same clear and almost linear downwind trend of decreasing particle size (Fig. 2). However, grain sizes in the seafloor sediments are substantially finer than found in the sediment-trap samples, and the downwind decrease in grain size is also less steep for the seafloor sediments. All traps show a "shoulder" towards the coarse end of the grain-size distribution, which is most prominent at station M5 (Fig. 3A). Such shoulders are also found in the seafloor sediments (Fig. 3B), showing that coarse particles are not only deposited at proximal locations, but also transported over great distances. These include

"giant" particles of up to 100 µm, which are observed as far west as station M4 (49° W; approximately 3500 km from the African coast), and consist of both platy mica and rounded quartz particles (Fig. 4). Furthermore, the average grain-size distributions show that the differences between stations are larger than between the upper (1200 m) and lower (3500 m) traps at stations M2 and M4 (Fig. 3A). The grain-size distributions of the seafloor sediments show that the dust at station M1 is the least sorted, meaning that the widest range of particles of different sizes is deposited closest to the source (Fig. 3B).

### 3.2 Seasonal grain-size trends

The particle size of Saharan dust deposited in the Atlantic Ocean changes seasonally, and is clearly coarser in summer than in winter at station M1 (Fig. 5A). During spring, a coarse shoulder is present in the grain-size distributions (Fig. 5B), which is more prominent than during the other seasons. Also modal grain sizes illustrate this seasonality, varying between 13 and 17 µm from October 2012 to May 2013 (fall to spring), followed by a sharp increase to 33 µm in June 2013, and stays coarse

for the entire summer season (Fig. 6) at station M1. Grain sizes decrease again in late August, and keep decreasing throughout the fall of 2013. At M2 the modal particle size of the upper trap decreases from fall to winter, from 17 µm to about 10 µm, followed by an increase to around 16 µm in May, continuing into summer and fall 2013. At M4 particle sizes of the upper trap decrease from 11 µm in fall 2012 to 8 µm in mid-spring 2013, after which they increase to around 13 µm throughout summer and fall 2013.

Overall, the particle size at the three sites show the same seasonality, with coarser dust in summer and fall and finer dust in winter and spring (Fig. 6). However, the difference in particle size between these seasons is greatest at M1, close to the source (Fig. 2). Here, particles are also least sorted and have the widest range in particle size, which gradually decreases westward towards M5. However, seasonal trends in modal grain size are more pronounced in the three upper traps at 1200 m than in the four lower sediment traps at 3500 m (Fig. 7). In the lower traps, the modal particle size at the more northern

station M2 is slightly finer than at the more southern station M3 from fall 2012 to spring 2013, with the exception of two samples that show unusually high modal grain sizes (in November 2012 and April 2013, shown as "outliers" in Fig. 7). From summer 2013 onwards, the modal grain size of M2 and M3 converge, with synchronous fluctuations between 15 and 19 µm.



Seasonality at M4 is even weaker, with grain sizes varying between 6 and 18 µm. At the westernmost station M5 modal particle size ranges between 4 and 10 µm, with a decrease in spring 2013 and an increase in summer. In all seven traps, dust is finest during spring. When comparing modal grain sizes found in the upper (1200 m) and lower (3500 m) traps from stations M2 and M4, it shows that the lower sediment traps have slightly coarser dust than the upper traps (Fig. 8).

## 4 Discussion

The grain size of dust decreases with increased distance from the source (Glaccum and Prospero, 1980;Goudie and Middleton, 2006;Mahowald et al., 2014;Stuut et al., 2005): coarse particles have a higher settling velocity and smaller particles can be transported over greater distances (Gillette, 1979;Tsoar and Pye, 1987). This mechanism accounts for the downwind fining observed in both the sediment traps and the seafloor sediments along the trans-Atlantic transect (Fig. 2). Since the seafloor sediments represent a longer time average of Saharan dust deposition than the sediment-trap samples, it implies that the downwind fining is a long-lived trend. Mahowald et al. (2014) hypothesize that dust in the high atmosphere is finer grained than in the lower atmosphere, which is in turn finer than the deposited dust, due to the preferential settling of coarse particles. However, we observed giant particles ($\geq$100 µm) as far as station M4 (49° W; approximately 3500 km from African coast) (Fig. 4). Most of these particles are mica particles, whose platy shape allows for aerial transportation over greater distances (Stuut et al., 2005). Indeed, the coarse particles causing the secondary peak in the grain-size distributions appear to increase downwind from M3 to M5 (Fig. 3), confirming that the coarse but platy mica particles are preferentially transported over greater distances. Such coarse particles are generally not incorporated into climate models (Kok, 2011). This underestimation of the coarse size fraction may have its origin in the sampling of dust of specific size classes, e.g. $PM_{10}$ and $PM_{2.5}$, which form the basis of the guidelines from the World Health Organization (WHO, 2006) on fine-grained particles.

The modal particle size of the sediment-trap samples is substantially coarser than that of the seafloor sediments at the same stations along the transect. The particle-size distribution found in the sediment-trap samples closely resemble Saharan dust sampled directly from the atmosphere, which has modal grain sizes varying between 8 and 42 µm (Stuut et al., 2005). By contrast, modal grain sizes in the underlying seafloor sediments range between 4 and 6 µm. Since the seafloor sediments represent a longer time period, this suggests that Saharan dust was significantly finer in the recent past than it is today. Deposition of coarser dust could be related to increased emission as a result of human activity since the nineteenth century due to commercial agriculture (Mulitza et al., 2010). Not only does that increase dust emissions, it also enables larger particles to be emitted (McTainsh et al., 1997), causing the particle size of Saharan dust to become gradually coarser over time, as we see now in the sediment traps.





The seasonal variability in particle size can be the result of several factors. First, it could result from the seasonal movement of the dust cloud, associated with the latitudinal movement of the ITCZ (Nicholson, 2000). As a result, in summer dust is transported at more northern latitudes than in winter, as indicated by the aerosol optical depth (AOD) data (Fig. 9). These aerosols can include sea salts, organic and black carbon, sulfates and mineral dust. However, the aerosols over our study area

are mostly mineral dust originating from the African continent (Yu et al., 2015). In summer, when AOD values are highest, the cloud is located at its northernmost position (Fig. 9D). Aerosol concentrations are lowest during fall (Fig. 9A and -E), and during winter the cloud is located in its southernmost position (Fig. 9B). However, during winter the aerosols may receive a higher contribution from soot by bushfires released more to the south (as also visible during the other seasons), thereby moving this high-AOD cloud southward and possibly falsely implying the latitudinal movement of the dust cloud.

This seasonal, latitudinal shift of the dust cloud is reflected in the samples from stations M2 and M3, which are positioned at one degree northern latitude from each other. During winter, modal grain sizes at the northern station (M2, 13˚ N) are finer than at the southern station (M3, 12˚ N), while similar at both stations during summer (Fig. 7). Thus, during winter the northern station M2 does not receive the same dust as station M3, since the dust cloud is located more to the south. In

summer the dust cloud is located more to the north, delivering coarser particles and at the latitude of both stations. However, the difference in grain size between the two traps is small, due to the close proximity of the two stations (about 200 km). In addition, the seasonal shift of the ITCZ also causes a latitudinal shift of the seasonal rain belt, affecting different sources during the year (Nicholson, 2000) and changing the amount and location of wet deposition. An alternative explanation is provided by different wind systems that are active throughout the year, along different trajectories and at different wind

speeds. These can entrain dust from different source areas. The elevation of these wind systems, in combination with wind speeds and the particle size of the source soils, determine the particle-size distributions of the entrained dust (Marticorena, 2014;Tsoar and Pye, 1987), and are further influenced during transport and deposition, creating different grain-size signatures for summer and winter dust.

In winter, dust is transported at lower altitudes than during summer. This is evidenced by satellite images of the Cape Verde

islands, which show the high mountain tops (highest point is Fogo at 2829 m) piercing through the dust cloud, deflecting it around the islands (Fig. 10A). The lowest peak that is still visible above the dust cloud is Brava (976 m), but the top of São Vicente (750 m) is not. This means that the top of the dust cloud is at an elevation between 976 and 750 m. In summer, dust is transported at much higher altitudes than winter, covering the Cape Verde islands in a thick blanket of dust (Fig. 10B), meaning that the top of the cloud is at an elevation of at least 2829 m. During summer, dust is transported in the high-altitude

Saharan Air Layer (SAL) (Carlson and Prospero, 1972;Kanitz et al., 2014;Prospero and Carlson, 1972;Tsamalis et al., 2013). Mahowald et al. (2014) argue that the dust particle size is most dependent on wind speeds at emission, and high wind velocities during summer would elevate coarse particles to the SAL. The wind in this air layer has velocities $>7$ ms$^{-1}$ (Tsamalis et al., 2013). This enables the transport of coarser dust particles in summer, and due to the high altitude these coarse particles are transported over great distances.



Four-day backward trajectories of air parcels also illustrate the difference in the elevation of the dust-transporting air layers between winter and summer (Fig. 10C and -D). The heights of these air layers were chosen in accordance with the hypothesized heights of the dust-carrying air layers, as demonstrated in Figures 10A and -B, with the lowest (500 m) air

layer representing winter dust transport and the highest (3500 m) air layer representing summer dust. In winter (Fig. 10C), the higher air parcel is not originating from the African continent, and therefore unable to transport dust to the sample location. The lower air parcel has a more eastern origin, and could be transporting dust (Fig. 10A), picked up from the surface and brought to higher altitudes. By contrast, in summer (Fig. 10D) the higher air parcel has a more continental origin and is the most likely dust-carrying air layer over the lower air parcel. The elevation profile shows that this high-elevation air

parcel started at lower altitudes, but upon reaching the coastline it was uplifted to about 3500 m AGL (Fig. 10D, bottom panel). This is in accordance with how the Saharan Air Layer (SAL) is described, when dust-carrying air from the continent is uplifted by a cool marine inversion layer (Carlson and Prospero, 1972;Prospero and Carlson, 1972). This inverted air layer is visible in the 500 m air layer, moving in an opposite direction, from west to east. After this sharp increase in altitude, the air layer decreases in altitude, which persists across the Atlantic Ocean (Tsamalis et al., 2013).

The summer season is also characterized by an increased number of more intense dust storms. From May to September, dust is almost continuously emitted from the African continent, as shown by satellite images (MODIS Terra and Aqua satellites; NASA Worldview). Within five days, the dust cloud propagates towards the Caribbean and becomes progressively thinner by dust deposition in the Atlantic Ocean along its track. Increased deposition of coarse particles can also be caused by increased precipitation in summer and fall, as opposed to almost no precipitation in winter and spring (Fig. 11). Increased

precipitation at station M1 seems to coincide with increased modal grain sizes, and this relation commences with lowest precipitation early June 2013. This suggests that little precipitation is already sufficient to wash out the suspended dust from the atmosphere by wet deposition.

At M1, the percentage of sand-sized particles (> 63 µm) increases sharply in spring while modal grain sizes increase in summer (Fig. 11). This increase in coarse particles is related to coarse shoulders in the grain-size distributions of the spring

samples (Fig. 5B) that are absent or less prominent in fall, winter and summer (Fig. 5A and -B). These coarse particles mostly include micas: due to their platy shape, these particles have a different aerodynamical behavior than more spherical quartz particles and are therefore more easily transported by wind than spherical particles with a similar diameter (Stuut et al., 2005). However, also large (≥ 100 µm) more spherical particles were observed in the samples, at very large distances from the source (Fig. 4). These coarse particles, visible in the grain-size distributions as coarse shoulders, are found in all the

traps at all stations, and appear most frequent during spring. More coarse particles during spring could mean that the dust could originate from a different source area. However, backward trajectories calculated over the entire sampling period do not suggest this.





The lower (3500 m) traps show less seasonality and are generally slightly coarser than the upper (1200 m) traps. This may be due to the disaggregation of marine snow, releasing the individual dust particles and thus decreasing their settling velocity. Therefore, it would take longer for particles to reach the lower traps at 3500 m, especially very fine particles, and as a result the particle-size distributions lose their seasonal characteristics. This would also explain why the dust in the lower traps (at M2 and M4) is slightly coarser than their upper counterparts, since these coarse particles settle more quickly, and the very fine particles may not reach the lower traps.

## 5 Conclusions

We have shown seasonal and spatial changes in Saharan mineral dust transport and deposition across the Atlantic Ocean by means of sediment-trap sampling between October 2012 and November 2013, and seafloor sediments at the same stations. Our results show strong seasonal variations and significant fining in particle size with increasing distance from the source, with modal grain sizes ranging from 4 to 33 µm. Coarser dust found in the sediment traps opposed to the seafloor sediments could result from emission of coarser dust due to the onset of commercial agriculture in the 19[th] century. The down-wind decreasing particle size reflects the greater gravitational settling velocity of coarse particles, resulting in deposition closer to the source. The largest seasonal difference in particle size occurs closest to the source, however the lower sediment traps (3500 m) show less seasonality than the upper sediment traps (1200 m). This may be due to marine snow disaggregating, decreasing the settling velocity of individual dust particles, resulting in a decreased expression of the seasonal particle-size signatures. Coarser grain sizes during summer and finer during winter and spring suggest: (1) summer transport at higher elevations of up to 5 km within the Saharan Air Layer at high wind speeds ($> 7$ ms$^{-1}$), compared to winter transport; (2) coupling to the latitudinal movement of the dust cloud with the ITCZ; (3) increased emission by more frequent dust storms in summer combined with wet deposition by increased precipitation. Increased contribution of coarse ($> 63$ µm) particles in spring is likely caused by large platy minerals (e.g. micas) of small aerodynamic size that are easily uplifted and transported, possibly from a different source area. These coarse particles are transported thousands of kilometers away from the Saharan source. Multiple-year samples from this transect should clarify which of the above mentioned processes are more dominant, in order to be applied in e.g. climate models and climate reconstructions. Our results contribute to a better understanding of the seasonal and spatial variability of Saharan dust, which still remains a poorly constrained factor in global climate.

## Acknowledgements

We thank the captain, crew and scientists of RV *Meteor* cruise M89 and RV *Pelagia* cruise 64PE378, for deployment and retrieval of the sediment-trap moorings. Funding is provided by NWO for the TRAFFIC project 822.01.008, as well as by the ERC with starting grant 311152: DUSTTRAFFIC.



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



**Table 1. Locations and depths of the sampling stations M1–M5. BSL = below sea level.**

| Station | Latitude (° N) | Longitude (° W) | Trap depths (m BSL) | Bottom depth (m BSL) | Distance to African coast (km) |
|---------|---------------|-----------------|---------------------|----------------------|-------------------------------|
| M1 | 12.00 | 23.00 | 1150 | 5000 | 700 |
| M2 | 13.81 | 37.82 | 1235, 3490 | 4790 | 2300 |
| M3 | 12.39 | 38.63 | 3540 | 4640 | 2400 |
| M4 | 12.06 | 49.19 | 1130, 3370 | 4670 | 3500 |
| M5 | 12.02 | 57.04 | 3520 | 4400 | 4400 |



**Figure 1. A: Map with sampling stations M1–M5 in the Atlantic Ocean at 12° N. B: Bathymetry along 12 N (from www.gebco.net). with sediment traps at 1200m and 3500m BSL. Crossed-out sediment traps could not be recovered.**




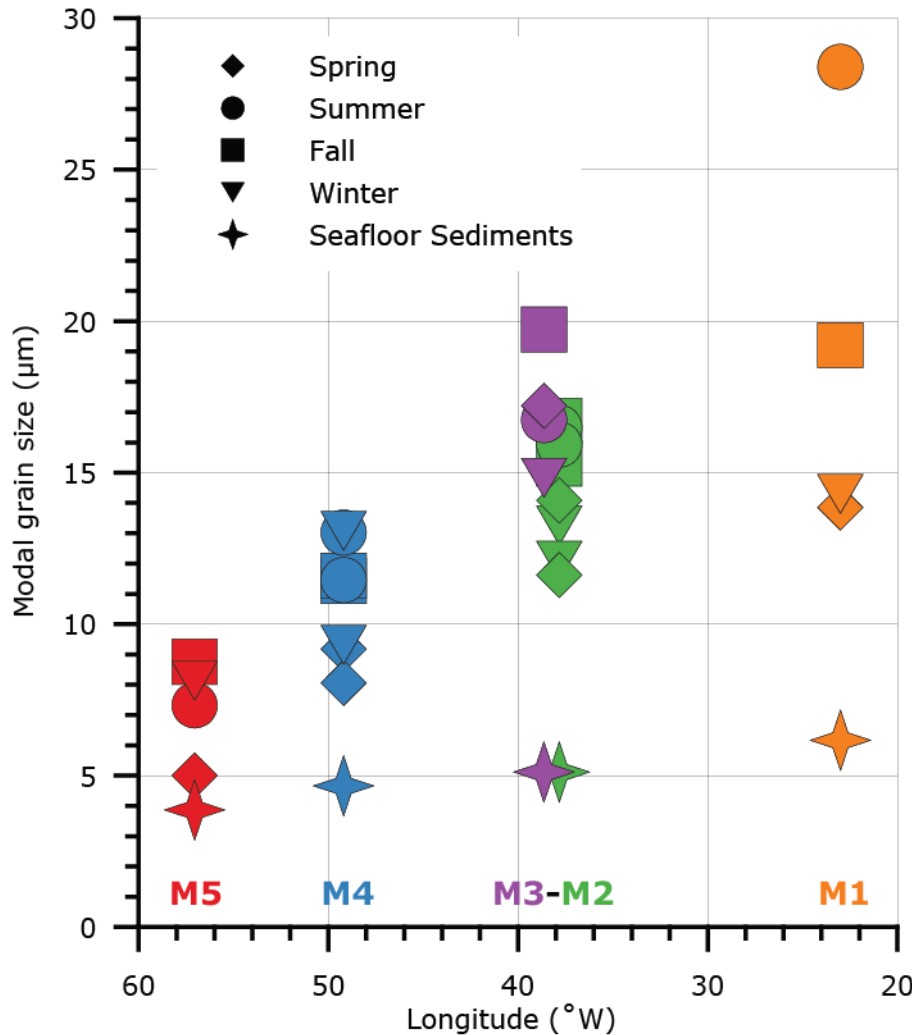

**Figure 2. Downwind fining and seasonality in average modal grain size per season for all seven traps, and modal grain size of the seafloor sediments, versus western longitude, for October 2012 –November 2013.**





**Figure 3. A: Average grain-size distributions of all seven sediment traps, representing the average of 24 samples, where U = upper trap (1200 m) and L = lower trap (3500 m). Collected between October 2012 and November 2013. B: Grain-size distributions of seafloor sediments at the five mooring stations (M1–M5) along the trans-Atlantic transect.**



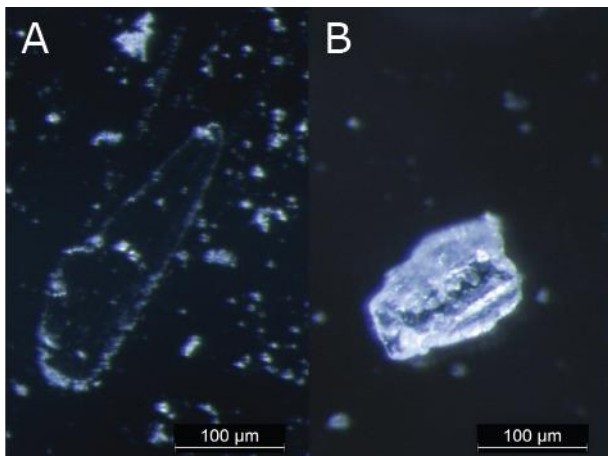

**Figure 4. Large dust particles from the lower (3500 m) trap at station M4 (12˚ N, 49˚ W; approximately 3500 km from African coast). A: Large mica particle (diameter approximately 250 µm over long axis) from sample 6 (January 7 – 23, 2013). B: Large quartz particle (diameter approximately 140 µm over long axis) from sample 22 (September 20–October 6, 2013).**



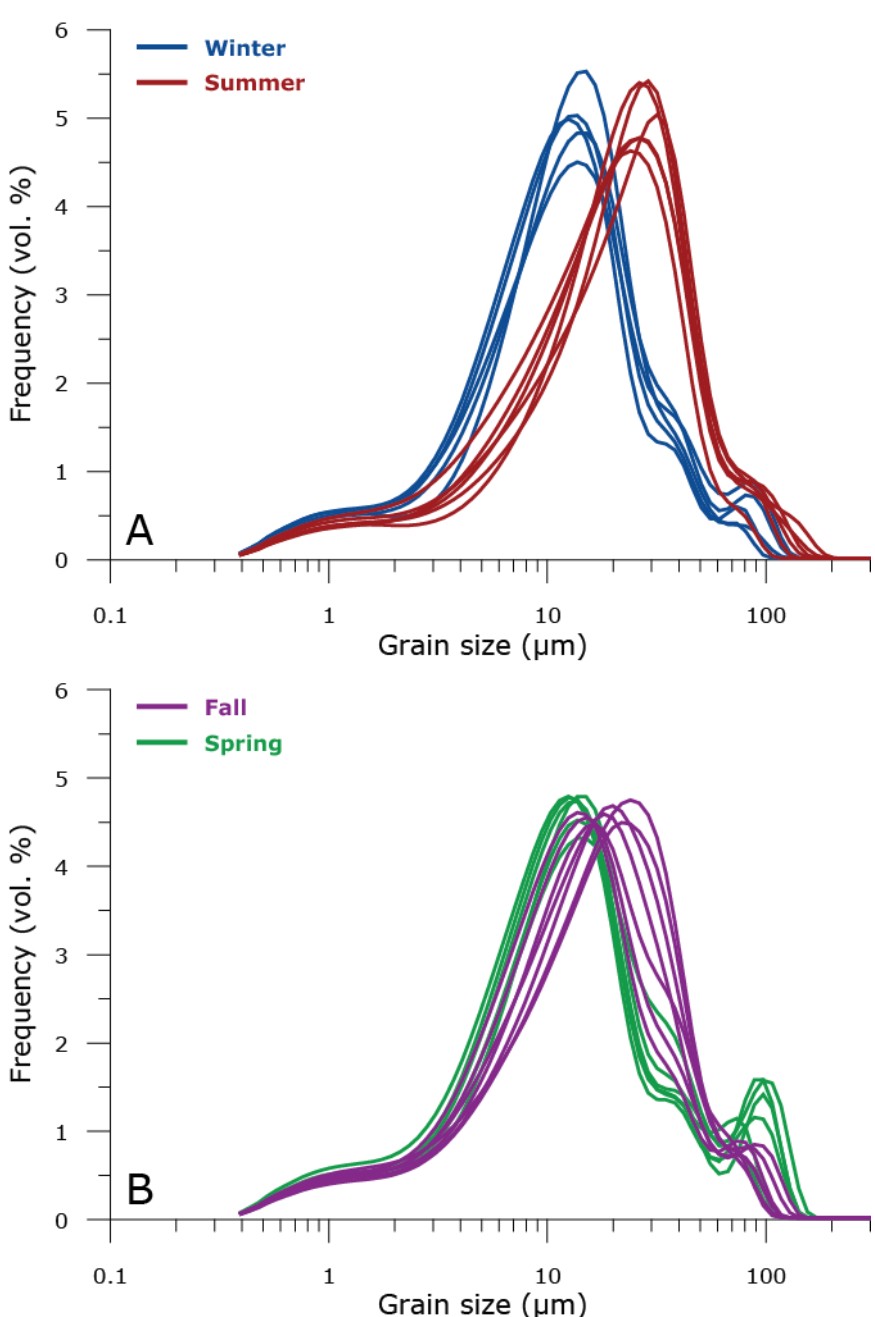

**Figure 5. Seasonal grain-size distributions at station M1 (12° N, 23° W; approximately 700 km from the African coast) for October 2012–November 2013, of A: winter (blue), summer (red), B: fall (purple) and spring (green).**





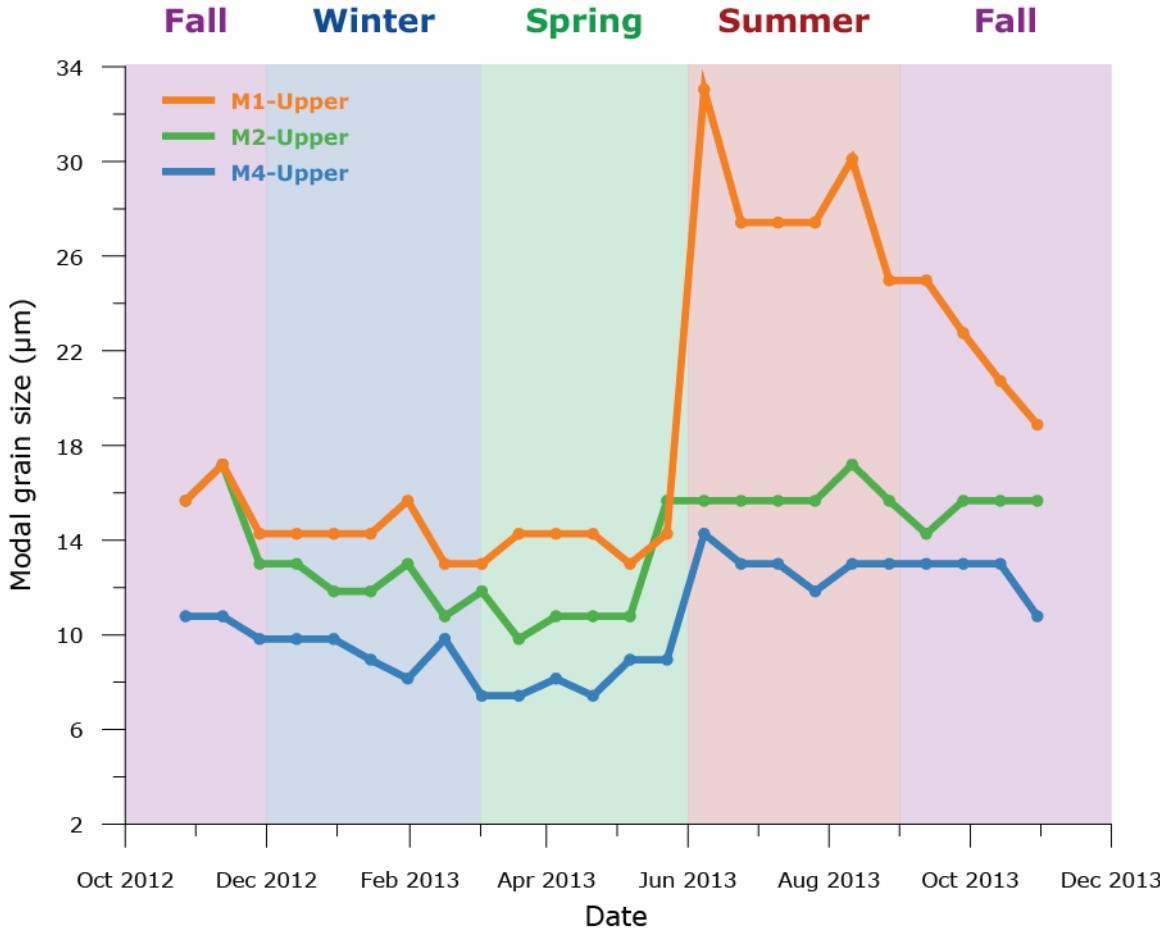

**Figure 6. Modal particle size of dust collected by the three upper (1200m) sediment traps at stations M1, M2 and M4, from October 2012–November 2013.**



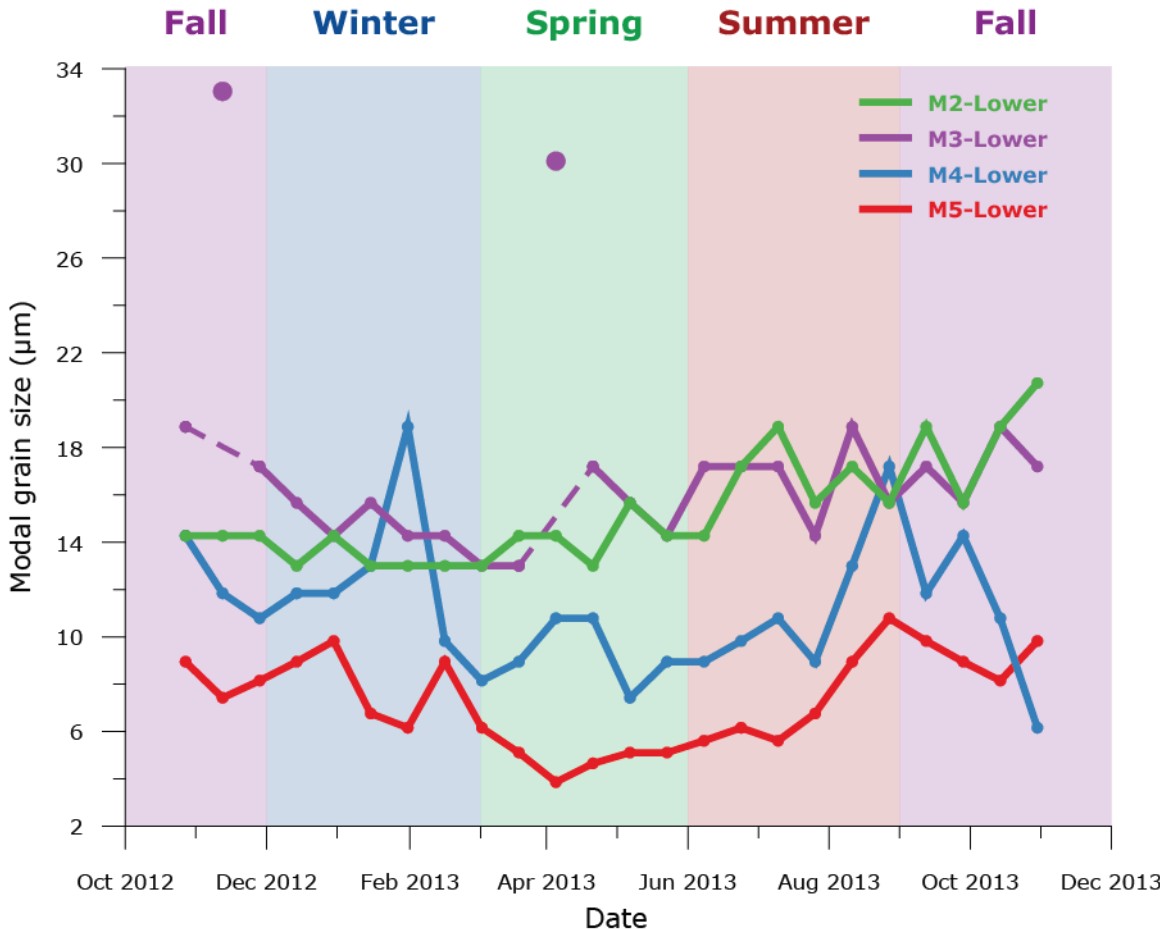

**Figure 7. Modal particle size of dust samples from the four lower (3500m) sediment traps at stations M2, M3, M4 and M5, for October 2012–November 2013. The two points that are not connected in series M3-Lower are considered outliers.**




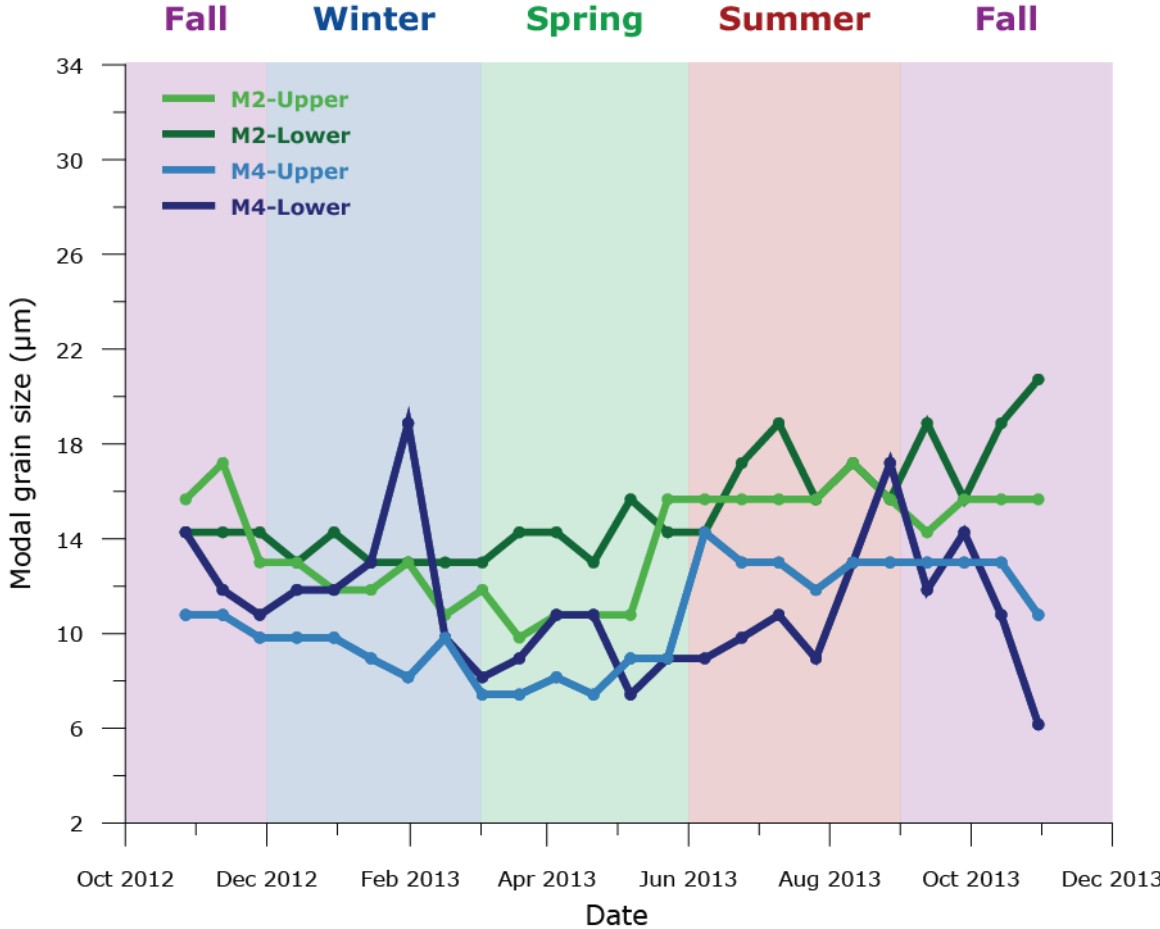

**Figure 8. Modal particle size of dust samples from the upper (1200 m) and lower (3500m) sediment traps at stations M2 and M4, for October 2012–November 2013.**





**Figure 9. Three-month average aerosol optical depth (AOD) for the sampled seasons, A: Fall (SON) 2012, B: Winter 2012–2013 (DJF), C: Spring 2013 (MAM), D: Summer 2013 (JJA) and E: Fall 2013 (SON). Stations M1–M5 are marked with black/white circles. Maps visualized with the Giovanni online data system, NASA GES DISC.**



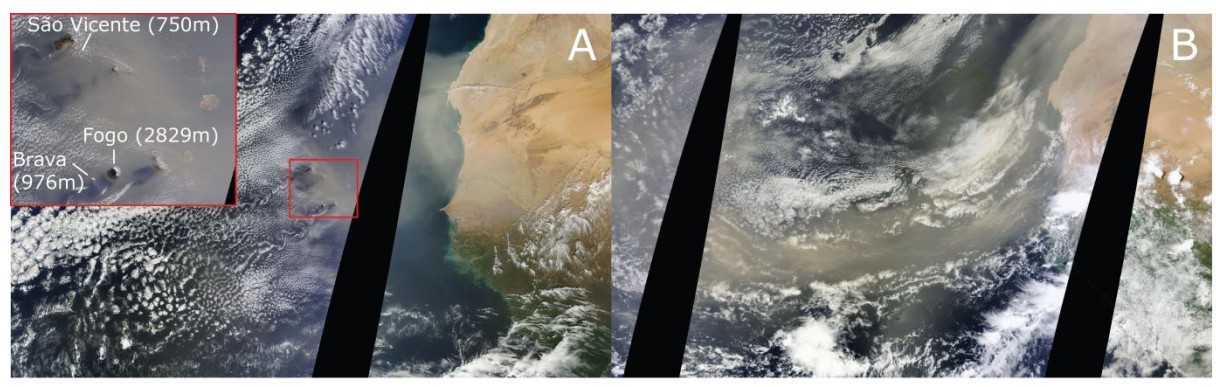

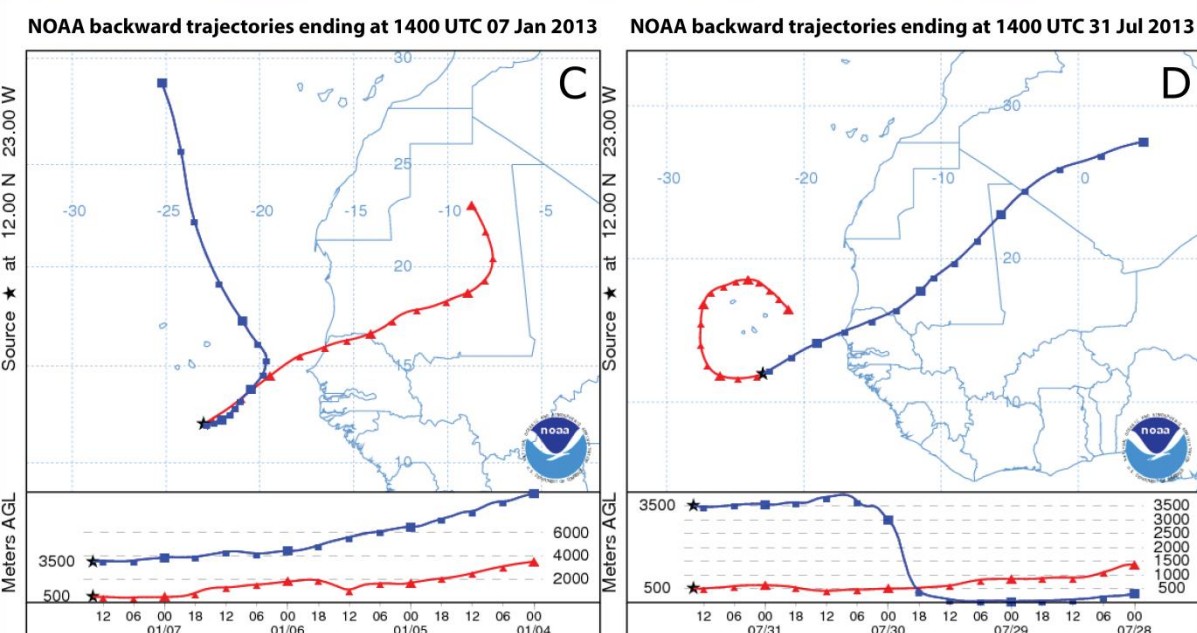

**Figure 10. A:** Satellite images of typical winter dust transport, with close-up of the Cape Verde islands (7 January 2013) and **B:** typical summer dust transport (31 July 2013) over the Cape Verde islands, at relatively low and high altitudes, respectively. Images from NASA Worldview, MODIS Terra satellite. Black areas are artefacts from satellite passage. **C & D:** Concurrent four-day backward trajectories of air parcels from station M1 (star), at 500 m (red) and 3500 m (blue) AGL, showing trajectory maps (top) and elevation profiles (bottom). **C:** ending at 7 January 2013, **D:** ending at 31 July 2013.



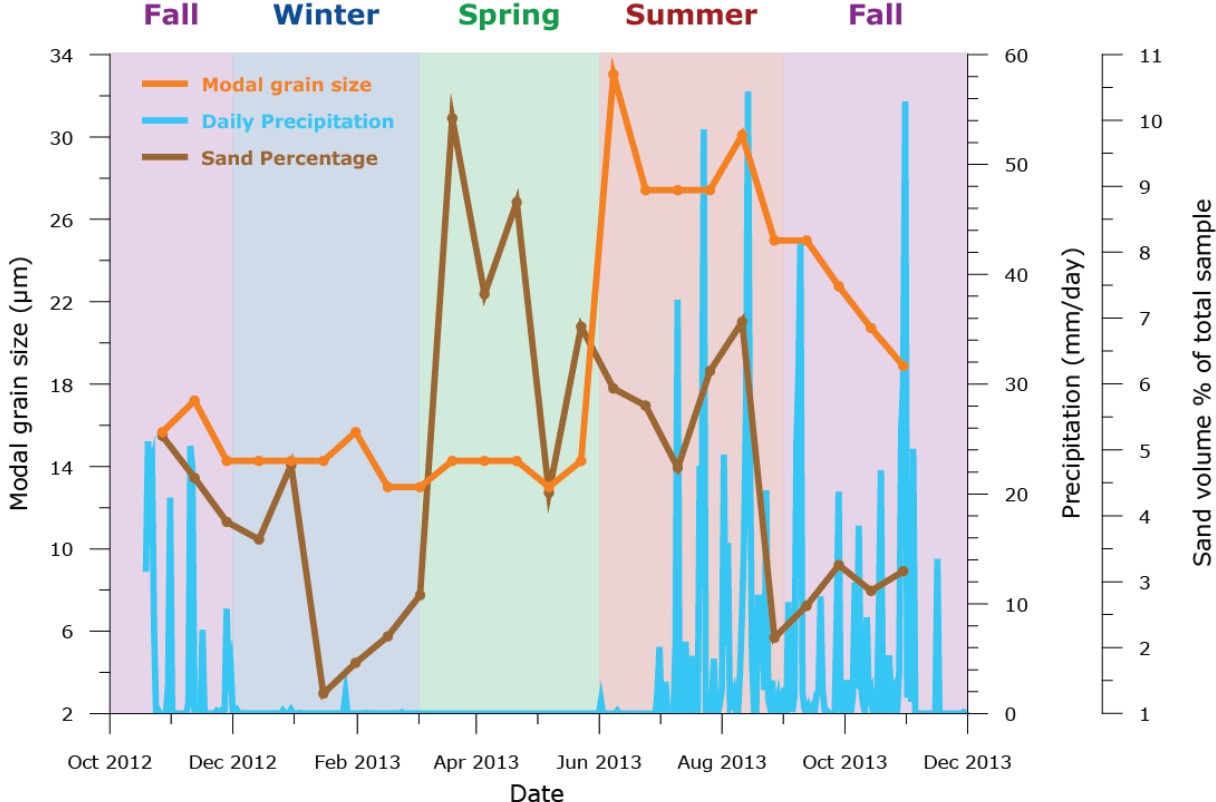

**Figure 11. Modal grain size (left axis), daily precipitation (right axis; from Giovanni online data system, NASA GES DISC) and percentage of sand particles (> 63 μm) (far right axis) at station M1 (12° N, 23° W).**