# Peer review of "Particle size traces modern Saharan dust transport and deposition across the equatorial North Atlantic"

_Atmospheric Chemistry and Physics, 2016_

## Short Comment (SC1) · 2 May 2016

As a preface, this is not a peer review, just a short comment with some questions from an interested scientist, which would improve my understanding and hopefully the understanding of other readers.

I would like to begin this short comment by saying that my expertise is predominantly in remote sensing; we need optical models of aerosols to use in our retrieval algorithms, because the measurements made from space are insufficiently to unambiguously determine all relevant parameters of interest. So, reports of mineral dust size distributions are of interest to me. I also know that terminology between the in situ and remote-sensing communities may differ. In that regard, I had a few questions about the dust

size distributions, which will help my understanding of these results in comparison with others:

1. The term "grain size" is used in multiple places. I was wondering how specifically the "size" is defined? Length of longest axis, diameter or radius of an equivalent spherical particle, something else? I did not spot that in the paper.

2. If I understand correctly, these grains were all recovered from moored traps within the sea itself at various depths. How closely are these expected to correspond to the size distribution of the dust in the atmosphere and falling on the surface? The paper (e.g. Figure 8) suggests that between the upper and lower sampling depth, there can be a difference of order a micron or so between modal sizes. Are the upper traps thought to be representative of dust in the atmosphere?

3. When calculating optical properties, due to the complex shapes of mineral dust grains, we typically model them as mixtures of ellipsoids of various sizes and with a distribution of different particle aspect ratios. The shape affects their scattering/absorption properties. I was wondering if you had any done any analyses of the particle shapes as well as the sizes?

---

## Author Comment (AC1) · 18 May 2016

Dear Andrew Sayer,

Thank you for your interest in our paper and your comments. We hope we answered your questions adequately.

1. The term "grain size" is used in multiple places. I was wondering how specifically the "size" is defined? Length of longest axis, diameter or radius of an equivalent spherical particle, something else? I did not spot that in the paper.

The particle sizes were determined with a Beckmann-Coulter laser particle sizer LS13320 and are expressed as equivalent-sphere diameter. The grain-size distribu-
tions are determined by the volume percentage of equivalent spherical particles of that size. Particle-size distributions consist of 92 logarithmic size bins, ranging from 0.375 to 2000 $\mu$m.

2. If I understand correctly, these grains were all recovered from moored traps within the sea itself at various depths. How closely are these expected to correspond to the size distribution of the dust in the atmosphere and falling on the surface? The paper (e.g. Figure 8) suggests that between the upper and lower sampling depth, there can be a difference of order a micron or so between modal sizes. Are the upper traps thought to be representative of dust in the atmosphere?

The samples from the sediment traps are indeed thought to be representative of dust travelling through- and settling from the atmosphere. We compared the dust from the traps to dust sampled from the air on filters with dust-collectors onboard ships during our research cruises, and these showed to be very similar to what we collect in the sediment traps and on the ocean seafloor. During one of these cruises (in 2015) we were very lucky to sail right through a dust event. Grain sizes from this event appeared smaller than what we found in the traps. However, this may also be the result of the fact that the dust sampled with these onboard dust-collectors is a relatively short snapshot in time ($\sim$24 hour sample), compared to the relatively longer sampling interval of the sediment traps (2-3 weeks). In addition, we added floating dust-collectors to the transect, at stations M3 and M4. These buoys collect dust on filters on a rotating carrousel similar to the sediment traps, sampling suspended dust directly from the atmosphere. We obtained the first successful filters from these buoys last month, but have not analyzed these yet. However, accumulative samples (1 year) of a passive air sampler (MWAC-sampler, see e.g., Goossens et al., 2000) attached to the buoys shows almost identical grain-size distributions to the sediment trap samples below, confirming again that the dust we find in the sediment traps is representative of the atmospheric dust.

3. When calculating optical properties, due to the complex shapes of mineral dust grains, we typically model them as mixtures of ellipsoids of various sizes and with a distribution of different particle aspect ratios. The shape affects their scattering/absorption properties. I was wondering if you had any done any analyses of the particle shapes as well as the sizes?

Particle-shape analysis was performed on other samples previously, but not yet for these samples. The analyses were done using a Sympatec QicPic, which works with a (2D) image analysis method and primarily measures the contour of a particle. The samples analyzed were from short sediment cores (multi-core sampler), recovered at the same stations M1-M5 of the transect. The the top centimeter was analyzed at every station, which showed the highest aspect ratio (highest symmetry of the particle) for the sediment at M1, closest to the African source, which was also the coarsest sample. The samples from M2-M5 showed lower, but similar to each other, aspect ratios. This could mean that close to the source, more (and more symmetrical) quartz particles are deposited, and at greater distances platy (clay) minerals with low symmetry are deposited. From the grain-size distributions as shown in the paper we could already see that more platy mica particles are deposited predominantly in spring. Furthermore, within each of the seafloor samples, it showed that the coarser particles have a lower aspect ratio than smaller particles, meaning that larger particles are less symmetrical than smaller particles.

With kind regards, Michèlle van der Does and co-authors

Cited reference Goossens, D., Offer, Z., London, G., 2000. Wind tunnel and field calibration of five aeolian sand traps. Geomorphology 35, 233-252.
* * *

---

## Referee Comment (RC1) · Anonymous Referee #1 · 2 Jun 2016

There is a lot of potential in this paper, but as written there are some serious issues with relating the data presented here with the conclusions. Even for big picture ideas (size gets smaller as you go across the Atlantic), it's not clear to me how to interpret particles from sediment traps, so I think this needs to be discussed much more explicitly in the paper. The data itself, with the size changes across the ocean, should be publishable, but it's the interpretation that is really an issue in this paper.

The big issue is: what is the aerosol in the trap and how is it related to what comes in at the top? Previous studies have shown that there are at least seasonally modulated relationships between the two (Bory et al., 2002), but here the authors are trying to

interpret these seasonal changes as occurring in the atmosphere, which could be true. But the fact that the aerosol size is systematically different at the deeper cores suggests there is something else going, and the assertion that this must be from changes in the dust sources (agriculture!) is too speculative to be convincing. I am not even sure I believe the sizes they are getting represent aerosols, and definitely the time and space lag issues related to aerosol transport and processing in the oceans is too important, and almost completely neglected here, will heavily modify the signal they are trying to interpret!

1. size: "This resulted in particle-size distributions consisting of 92 logarithmic size classes ranging from 0.375 to 2000 $\mu$m. Grain-size statistics were calculated geometrically using the graphical method of Folk and Ward (1957) using GRADISTAT (Blott and Pye, 2001)." It sounds like you are assuming that the size of the particles you are measuring in the sediment trap is the size of the particles in the atmosphere? There is a lot that goes into that kind of set of assumptions, so please spend at least a paragraph in the methods describing why you think this will work, previous papers which showed a relationship (or not) and what kind of assumptions it requires. Wetting an atmospheric aerosol during deposition, could either make particles coagulate or break the bonds of particles. On the other hand, material can coagulate onto the particles in the ocean, and change or process them. Already your evidence that the sizes are different at the trap and deeper down suggests changes in size or processing (or advection: see next point). Please be explicit about the assumptions you are making and justify them here in the results, and then in summary and conclusions discuss the implications of your assumptions for your work. Are these aerodynamic or geometric measurements, as a basis? Size of aerosols is tricky to measure (e.g. Reid et al., 2003) and different measurement methods get quite different results: how do your methods compare? Is there any way you can use previous measurements of size in aerosols (e.g. Reid et al., 2003; Skonieczny et al. (2013)) to help you with this problem? How do you know that these particles are from eolian deposition and not some other process?

2. advection and sedimentation rates: The second issue is the relationship of what enters the top of the ocean and what is deposited in the sediment traps. Previous studies (Bory et al., 2002) have suggested that productivity could modulate the transfer rate between the top of the ocean, and the cores. We know that the dust has to be carried with the current as it floats downward: how far downward? (Han et al., 2008; Siegel and Deuser, 1997) show that it really can be quite far. If it also seasonally being modulated, that would really mess up your signal!

3. deposition rates. What are the deposition rates you are getting? Are they consistent with your assumptions? Are the deposition rates reasonable? Are they the same in the sediment traps as the sediment below? Please describe this a bit more.

More details

"However, grain sizes in the seafloor sediments are substantially finer than found in the sediment-trap samples, and the downwind decrease in grain size is also less steep for the seafloor sediments." What does this mean for interpretation? The more processed the cores, the finer they look? Or that they are being dissolved? Or that they are advected from farther upstream? I find this observation very difficult to understand, and makes me doubt your methodology.

"Since the seafloor sediments represent a longer time average of Saharan dust deposition than the sediment-trap samples, it implies that the downwind fining is a long-lived trend." How long is the time average for the sedments on the seafloor compared to the traps?

"Mahowald et al. (2014) hypothesize that dust in the high atmosphere is finer grained than in the lower atmosphere, which is in turn finer than the deposited dust, due to the preferential settling of coarse particles. However, we observed giant particles ($\geq$100 $\mu$m) as far as station M4 (49ËŽ W; approximately 3500 km from African coast) (Fig. 4)." On the surface of it, these two statements have nothing to do with each other, since one is talking about vertical height in the atmosphere and the other is talking

about horizontal distance from Africa. You seem to be implying that they are somehow contradicting each other, but it doesn't seem possible to infer from distance downwind anything about vertical structure tof the atmosphere?.

"The particle-size distribution found in the sediment-trap samples closely resemble Saharan dust sampled directly from the atmosphere, which has modal grain sizes varying between 8 and 42 $\mu$m (Stuut et al., 2005)." This is really important, but you don't say where this observation is made? Size is varying along the transect in the atmosphere also. . .. what type of observation is this? What kind of uncertainties are in that method (i.e. look at (Reid et al., 2003))

"By contrast, modal grain sizes in the underlying seafloor sediments range between 4 and 6 $\mu$m. Since the seafloor sediments represent a longer time period, this suggests that Saharan dust was significantly finer in the recent past than it is today." And in the conclusions: "Coarser dust found in the sediment traps opposed to the seafloor sediments could result from emission of coarser dust due to the onset of commercial agriculture in the 19th century." This is a huge jump, which seems incredibly unlikely. Most likely there is ocean processing. . ..

"The lower (3500 m) traps show less seasonality and are generally slightly coarser than the upper (1200 m) traps. This may be due to the disaggregation of marine snow, releasing the individual dust particles and thus decreasing their settling velocity. Therefore, it would take longer for particles to reach the lower traps at 3500 m, especially very fine particles, and as a result the particle-size distributions lose their seasonal characteristics. This would also explain why the dust in the lower traps (at M2 and M4) is slightly coarser than their upper counterparts, since these coarse particles settle more quickly, and the very fine particles may not reach the lower traps." This is really important, and should be talked about first: you need to convince us that you can say anything about seasonality in the dust size from sediment trap data, especially with the observed bias between sediment trap and core sizes. So I would start from this and really convince us that any of the signal is actually from the atmosphere first.

“We have shown seasonal and spatial changes in Saharan mineral dust transport and deposition across the Atlantic Ocean by means of sediment-trap sampling between October 2012 and November 2013, and seafloor sediments at the same stations.” So at this point, this statement has not been proven: you have only shown sea floor sediment changes in size. It is interesting that you see these trends, but anything about the atmospheric aerosols is speculation.

Han, Q., Moore, J.K., Zender, C., Measures, C., Hydes, D., 2008. Constraining oceanic dust depostion using surface ocean dissolved Al. Global Biogeochemical Cycles 22, doi:10.1029/2007GB002975. Reid, J.S., Jonson, H., Maring, H., Smirnov, A., Savoie, D., Cliff, S., Reid, E., Livingston, J., Meier, M., Dubovik, O., Tsay, S.-C., 2003. Comparison of size and morphological measurements of dust particles from Africa. Journal of Geophysical Research 108, 8593: doi:1029/2002JD002485. Siegel, D.A., Deuser, W.G., 1997. Trajectories of sinking particles in the Sargasso Sea: modeling of statistical funnels above deep-ocean sediment traps. Deep-Sea Research 44, 1519-1541.

---

## Referee Comment (RC2) · Anonymous Referee #2 · 3 Jun 2016

Van der Does and colleagues present their preliminary findings for one year of data from a multi-year sampling campaign, aimed at retrieving samples of dust deposited to the equatorial Atlantic Ocean (at latitudes ∼12 N), utilizing a transect of moored sediments traps at different water depths, downwind from the North African dust sources. The aim of this ambitious project to better constrain the evolution of the North African dust plume is of certain interest, and it is positive for the atmospheric and dust communities to be informed about these preliminary findings. This is a very interesting study, and the manuscript is in general well organized and quite clear. Nonetheless I do revise three major aspects that would deserve some revision, in order to clarify the work and perhaps improve the possible interpretation of the data.

General major comments

The particle size distributions are central in the manuscript. Nonetheless the descriptive metrics that are used for some of the diagnostic plots are only briefly mentioned. I think it would be very important to clearly show a validation of the metric used (e.g. mode of distributions fitted using GRADISTAT) against the specific observational data, before following with the discussion. This is particularly relevant since many of the samples show an apparent bi-modal distribution.

In the discussion of the data, the coarseness of grain size distributions is assumed to mimic the behavior of dust deposition flux, despite the fact that such information is not reported. In addition, a full comparison among different samples is hampered by the lack of this piece of information.

Some of the interpretations of the data provided in the discussion remain rather speculative. Please see the specific comments below.

Specific comments

2 / 8-9. Which is the mechanism that would explain this statement, in relation to the previous sentence?

3 / 26-32. I would argue that the point here is that we need to quantify how far a significant number of those particles can travel, to both (a) constrain the inputs to the ocean and (b) be able to estimate if, because of their actual amount, they are in fact relevant in terms of direct radiative effects or rather they could actually be ignored from this point of view, as typically done so far in models. That is why it would be important to have dust deposition fluxes associated to the size distribution data. If this piece of information is not available, the discussion should take into account this fact, and the interpretation of similarities / differences in the samples should be pondered accordingly.

4 / 30. Briefly, why did you discard some of the traps?

5 / 19-20. Please clarify if you refer to radius or diameter, here and throughout the manuscript.

5 / 20-21. As indicated above, please discuss much more extensively this aspect. For instance, describe how the method works, and show the comparison of the full distribution and the metric (mode) for two-three representative cases, e.g. a typical sample for each Winter, Summer, Spring from Figure 5. This should highlight how the metric vary according to the distribution's shape, thus help better understanding/constraining the following interpretations. This should also highlight whether the choice of the metric is the best option or better ones could be adopted in this case.

5 / 30. Did you only simulate four days? Later in the manuscript (9 / 31-32) it sounds like you may have selected four days out of a larger ensemble? Is that the case? If so, it would be interesting to see those. If not, how did you exactly determine that those would be representative days?

6 / 1. Do you have dust deposition flux data? I believe that all the comparisons among the samples in this study and the derived interpretations are subject to the limitation of not being associated to dust deposition fluxes. Therefore only partial information is available to derive conclusions.

7 / 14-17. As already mentioned, I think that the point is not whether a handful of giant particles make it a great distance, but rather how many and how far. If they appear to be quantitatively important, then this suggest that models should account for that, and they will need data to constrain their results. Hopefully your study will help addressing this issue!

7 / 17. "Preferentially" vs what? Please clarify this sentence.

7 / 25-30. This paragraph seems very speculative: there is no support to it in the discussion, and no time control is reported about the age of those seefloor sediments.

8 / 11-15. Here you seem to suggest a direct relation between coarse grain size and

high dust load (or AOD), and for extension to a high deposition flux? The reported study of Skonieczny et al. (2013) on the other hand shows coarser dust deposition at M'Bour, Senegal, associated with the season of low dust deposition flux. How would you justify your assumption in light of that? I think that absolute magnitudes of size distributions could help here in two different ways, most importantly with reference to Figures 6, 7, 8, 11. First, absolute values of particles concentrations (i.e. counting statistics on the direct output from the particle counter) may help to understand if the "shoulders" associated to the larger particles are actually statistically significant in all cases. One can see "tail effects" associated to sometimes individual large particles in low concentration samples such as from ice cores (e.g. Albani et al., 2012). This piece of information should be considered together with the choice of the mode as a metric to compare those samples. Second, even when samples are screened against possibly noisy signals, any interpretation on the actual quantitative transport potential (whether with season, or distance, or depth) of giant particles remains speculative without deposition flux data. The same way, in order to trace the spatial evolution of the North African dust plume, size distributions are necessary but not sufficient. Comparing sediment records from the Atlantic on different size ranges in fact yield surprising results, demonstrating the importance of considering both size distributions and fluxes (Albani et al., 2015). If this piece of information is missing, then the discussion should be extended to discuss the possible limitations of the derived interpretations.

8 / 24-28. Interesting approach!

8 / 31-32. Quite the opposite. I cite: "On balance, the measurements (Fig. 4) indicate that dust PSD is independent of the wind speed at emission. This conclusion is supported ..."

9 / 3. I would suggest changing "these air layers" with something like "the starting points for back-trajectories calculations".

9 / 8-9. This sentence is not very clear, please rephrase.

9 / 12-14. It seems that here "air-layer" is used to indicate "air parcel trajectory"?

9 / 15-18. You are not showing this. Please at least provide some reference.

9 / 18. "Increased deposition": where?

9 / 21-22. How do you know? You do not show any information about the atmospheric column above.

9 / 23-24. Again, it is not clear whether the mode is a good metric to compare bi-modal distributions.

9 / 25-30. How does a laser particle counter sees a flat particle? Overestimate it's spherical equivalent diameter? See e.g. Reid et al. (2003). How do you interpret this in your data, and according to the evolution of size distribution with distance from the source?

9 / 30-32. As already mentioned, if more back-trajectories calculations were performed, it would be interesting to see them.

10 / 1-6. Also in this respect, absolute values of concentration and most importantly dust deposition fluxes might shed some light on the issue. In addition, a little more discussion on the fate of particles throughout the water column and the the expected relation to the corresponding surface water and atmosphere could be added here.

10 / 11. Please add also here in the conclusions whether you refer to particle radius or diameter.

10 / 11-12. As indicated earlier, this statement is so far very speculative.

10 / 22-23. From your study, one would expect to learn how many.

Figure 2. Please differentiate the markers based on the depth for M2 and M4.

Figure 7. Could you provide a brief explanation about those outliers?

References

Skonieczny, C. et al.: A Three-Year Time Series of Mineral Dust Deposits on the West African Margin: Sedimentological and Geochemical Signatures and Implications for Interpretation of Marine Paleo-Dust Records, Earth and Planetary Science Letters, 364, 145-156, http://dx.doi.org/10.1016/j.epsl.2012.12.039, 2013.

Albani S., B. Delmonte, V. Maggi, C. Baroni, J.R. Petit, B. Stenni, C. Mazzola, and M. Frezzotti: Interpreting last glacial to Holocene dust changes at Talos Dome (East Antarctica): implications for atmospheric variations from regional to hemispheric scales, Clim. Past, 8, 741-750, doi: 10.5194/cp-8-741-2012, 2012.

Albani S. et al.: Twelve thousand years of dust: the Holocene global dust cycle constrained by natural archives, Clim. Past, 11, 869-903, doi:10.5194/cp-11-869-2015, 2015.

Reid, J. S., et al.: Comparison of size and morphological measurements of coarse mode dust particles from Africa, J. Geophys. Res., 108, 8593, doi:10.1029/2002JD002485, 2003.
* * *

---

## Referee Comment (RC3) · Anonymous Referee #3 · 8 Jun 2016

van der Does et al. present observations of what is thought to be Saharan dust along the trans-Atlantic transport pathway over the course of roughly a year. Their observations cover a wide lateral range across the Atlantic Ocean and demonstrate the decrease in particle size with increasing distance from the source. Additionally the paper is well written. Although these observations are interesting and worthy of representation in the literature, there are a few major concerns I express in the below review that should be addressed prior to final publication.

Major concerns

The assumption that the particles collected in the traps are all mineral dust from the Sahara seems like an over-interpretation of the results. First, there could still be inter-

ference from biological particles. The authors did carryout chemical degradation and deactivation techniques to denature biological constituents. However, these types of methods do not remove all of the viable cells; they simply kill them off while leaving behind a particle. They do not completely disintegrate under these methods. How did the authors account for leftover, dead cells or biological particles such as pollen or marine microorganisms, which can easily fall within the size range of what was measured? Second, there could also be contribution from sea salt particles or non-viable organic material from the ocean surface microlayer. How did the authors eliminate these other types of particles as potential candidates for what was sized? Third, the dearth of chemical or mineralogical analysis also forces me to question the conclusion that most of what was observed was dust. This could easily alleviate the issue by imaging and/or determining the composition of the particles in the samples. The authors do show one image of a dust particle, but was this conducted for all samples and multiple particles per sample? Maybe SEM/EDX, XRF, and/or XRD were conducted? Surely it may be too late to conduct such analyses, and if the authors decide to proceed with publication with the current methods only, should very clearly state the assumptions made regarding what the particles are and perhaps provide more background from previous work demonstrating dust observations in the Atlantic Ocean to support their assumptions. As a suggestion, it might be beneficial to look at salinity and surface chlorophyll concentrations of the domain over which the particles were transported to show possible sources of particles other than dust (or partially eliminate these as contributing sources).

How representative are the lower and sea floor traps of the observations of particle deposition and sedimentation during the study time period? Especially the sea floor, could these particles result from years of sedimentation and ocean circulation/currents introducing particles from all over the ocean system? It seems as if the ocean floor would be even more of a hodgepodge of all types of particles; this is where some sort of compositional information on the particles in the samples would be useful. Along these lines, I am not convinced that the smaller particles observed on the sea floor are

simply due to the fact that larger particle emission has occurred over time based on the methodology and observations presented.

Although it is generally understood that the SAL is transported westward over the Atlantic, the authors draw many conclusions of the seasonal altitude dependence of air mass transport and at only one trap location (M1). What would strengthen the argument regarding the impact of transport conditions and seasonal climate patterns on particle deposition/size is an ensemble or cluster analysis of HYSPLIT trajectories. The authors do state, "However, backward trajectories calculated over the entire sampling period do not suggest this. . ." which indicates that more trajectories were simulated. It would be helpful to show these to clearly show the seasonal variability. It would also be useful to conduct HYSPLIT analyses at all of the trap locations to better connect the sites and perhaps show that transport over the trap farthest from Africa does not experience as much transport as the trap closest.

General comments

The figures present data from a number of sources (i.e., MODIS and particle imaging). Although the captions to these figures briefly describe these data sources, they should be more comprehensively described in the methods section. As an example, what instrument was used to image the particles? How many images were acquired? Was this conducted for all samples? With respect to MODIS, provide at the very least a brief description of the satellite and how the data were acquired. For the precipitation, was this acquired from TRMM? Over what domain?

Specific comments

Page 2, line 19: Most people know what CALIPSO is, but do define the acronym.

Page 9, line 23: Only sand can be this size? What about large minerals? This seems like a vague definition without any measurements of the mineralogy.

---

## Editor Comment (EC1) · J. Schwarz (Editor) · 1 Aug 2016

Dear Dr. Van der Does and co-authors -

Thank you for your detailed response to the reviewers, and your improved manuscript. As you point out in the response, this manuscript represents a marriage across disciplines. Given that ACP is focused on the atmospheric issues addressed here, my sense is that a little bit more concession to the target community is called for.

For example, the widely raised question from the reviewers about potential biological sources of the lithogenic particles; I imagine that they were thinking (as I did) of the marine organisms that contribute their shells to limestone in the form of calcium carbonate. Could you more directly address this issue?

Other basic questions are still focused on the transport of dust from the surface downwards. Simple calculations of particle sedimentation rates (in the absence of any water currents) indicates that the short transport time between the 1200 m and 3500 m traps cannot be explained simply by particle setting. For example, for a 10 $\mu$m particle of density 2700 kg/m^3, a distance of only ~200 m would be traversed in a ~few weeks. Since 10 $\mu$m is close to the mass-modal diameter, it's clear that something else must be going on. Hence the interest in currents, I imagine, which the atmospheric community does not have any intuition about. Adding information about the rate and direction of currents will be very helpful. This is also relevant to the apparent discontinuity in results at the lower traps and at the ocean floor; why would transport be very speedy from 1200 m to 3000m, but then essentially disappear in the lowest 1 - 2 km?

Addressing these questions will help the atmospheric community contextualize the potential scale of any assumptions and uncertainties, and will help it grasp the significance of this manuscript as a step in achieving closure between atmospheric transport of mineral dust, and its removal from the atmosphere by dry and wet deposition.

As I had some concern that some portions of the reviewers' comments may not have been fully considered (these are the questions I reformed here; perhaps I am mistaken!), I have asked them, also, to comment on the response.

Thank you very much for working through this review process with ACPD; I have high expectations for the value of your technique and findings to contribute meaningfully to understanding and quantifying total and size-dependent dust transport and removal from the atmosphere.

PS - Minor additional comments:

It would be helpful to use clearer wording to describe the size distribution information throughout the paper. I found myself repeatedly questioning whether number or mass

average values were being presented. For example, in the caption of Figure 3, the "average modal grain size" is presented.I think this is the average modal-mass grain size ? ; in Figure 4 the caption is for "average grain-size distributions" , but these represent average grain-volume distributions? Page 19 ,lines 3-7 : "...with modal particle diameters ranging..." could perhaps be more clearly stated "... with mass-modal particle diameters ranging..."? And so on.

On Figure 4, I'd also suggest a clarification of the vertical axis legend. As it looks like the width of the bins of the histogram are not one size in linear space, but one size in log-space, it is standard practice in the atmospheric community to specify the volume fraction as scaled by the log-space bin width (dV/dLog(D), the differential volume per horizontal step in log space). I imagine that this is what is shown already.

Finally, my own elementary question reflecting lack of familiarity with your water-centric techniques: are the sampling volumes closed when not sampling/during recovery? Is there any chance that some vials sampled substantially longer times than others? If these are non-issues, that will be helpful to know.

---

## Author Comment (AC2) · 1 Aug 2016

Dear Dr. Schwarz,

Thank you for the time you spent assessing the manuscript. We would also like to thank the three anonymous reviewers for their comments. We considered every comment carefully and made changes to the revised manuscript accordingly.

We have found similar comments on the manuscript from different reviewers, and we are of the understanding that this is mainly due to the different "languages" researchers from different disciplines speak. One of the reviewers' main concerns deals with the representation of the lithogenic particles found in the sediment traps as aeolian dust.

As can be seen in the answers to the comments of the reviewers below, there are many reasons for this assumption. First, the sediment traps are located far away from any other possible source, e.g. riverine sediments, resuspension from shelf or bottom sediments, etc. Second, we compared grain size of the lithogenic particles collected by the sediment traps to dust collected from the atmosphere, directly above the sediment traps, which showed to be almost identical. Third, when comparing the upper (1200 m) and lower (3500 m) sediment traps from the same station, there are many indications of the nearly vertical settling of particles, indicating their atmospheric origin. And finally, when considering the large amounts of Saharan dust that are transported across the Atlantic Ocean every year, about 182 Tg (Yu et al., 2015), it will be fair to assume this will be a large contribution to the particle flux in the sediment traps.

Another question raised by the reviewers concerns the particle size of the dust we find in the traps, and if this is affected by some sort of processing or aggregation of particles. Aggregation of particles during or after deposition is possible, however all aggregates are destroyed by the three-step pretreatment processes to remove organic material in the sediments prior to grain-size analysis, including aggregates of dust particles that already existed during transportation in the atmosphere. Grain-size analysis is performed on the lithogenic fraction of the samples only. Therefore, the particle-size distributions of the dust measured in this study is at the very least an underestimation of the size of dust particles and aggregates as transported across the Atlantic Ocean.

With this paper we would like to bring the different disciplines together, which is also why we have chosen this journal for publishing. We greatly value the reviewers' comments and hope we have answered them adequately. Please find attached (in the supplement) the response to the reviewers' comments, including the revised manuscript with tracked-changes with respect to the original manuscript. For reviewer 1, the changes were made in red, for reviewer 2 in green, and for reviewer 3 in blue.

Please also note the supplement to this comment:

http://www.atmos-chem-phys-discuss.net/acp-2016-344/acp-2016-344-AC2-supplement.pdf

**Supplement:**

Dear Dr. Schwarz,

Thank you for the time you spent assessing the manuscript. We would also like to thank the three anonymous reviewers for their comments. We considered every comment carefully and made changes to the revised manuscript accordingly. For reviewer 1, the changes were made in red, for reviewer 2 in green, and for reviewer 3 in blue.

We have found similar comments on the manuscript from different reviewers, and we are of the understanding that this is mainly due to the different "languages" researchers from different disciplines speak. One of the reviewers' main concerns deals with the representation of the lithogenic particles found in the sediment traps as aeolian dust. As can be seen in the answers to the comments of the reviewers below, there are many reasons for this assumption. First, the sediment traps are located far away from any other possible source, e.g. riverine sediments, resuspension from shelf or bottom sediments, etc. Second, we compared grain size of the lithogenic particles collected by the sediment traps to dust collected from the atmosphere, directly above the sediment traps, which showed to be almost identical. Third, when comparing the upper (1200 m) and lower (3500 m) sediment traps from the same station, there are many indications of the nearly vertical settling of particles, indicating their atmospheric origin. And finally, when considering the large amounts of Saharan dust that are transported across the Atlantic Ocean every year, about 182 Tg (Yu et al., 2015), it will be fair to assume this will be a large contribution to the particle flux in the sediment traps.

Another question raised by the reviewers concerns the particle size of the dust we find in the traps, and if this is affected by some sort of processing or aggregation of particles. Aggregation of particles during or after deposition is possible, however all aggregates are destroyed by the three-step pretreatment processes to remove organic material in the sediments prior to grain-size analysis, including aggregates of dust particles that already existed during transportation in the atmosphere. Grain-size analysis is performed on the lithogenic fraction of the samples only. Therefore, the particle-size distributions of the dust measured in this study is at the very least an underestimation of the size of dust particles and aggregates as transported across the Atlantic Ocean.

With this paper we would like to bring the different disciplines together, which is also why we have chosen this journal for publishing. We greatly value the reviewers' comments and hope we have answered them adequately. Please find attached the revised manuscript, with tracked-changes with respect to the original manuscript.

**Anonymous Referee #1** (Referee comments in black, our reply and changes in the manuscript in red)

There is a lot of potential in this paper, but as written there are some serious issues with relating the data presented here with the conclusions. Even for big picture ideas (size gets smaller as you go across the Atlantic), it's not clear to me how to interpret particles from sediment traps, so I think this needs to be discussed much more explicitly in the paper. The data itself, with the size changes across the ocean, should be publishable, but it's the interpretation that is really an issue in this paper.

The big issue is: what is the aerosol in the trap and how is it related to what comes in at the top? Previous studies have shown that there are at least seasonally modulated relationships between the two (Bory et al., 2002), but here the authors are trying to interpret these seasonal changes as occurring in the atmosphere, which could be true. But the fact that the aerosol size is systematically different at the deeper cores suggests there is something else going, and the assertion that this must be from changes in the dust sources (agriculture!) is too speculative to be convincing. I am not even sure I believe the sizes they are getting represent aerosols, and definitely the time and space lag issues related to aerosol transport and processing in the oceans is too important, and almost completely neglected here, will heavily modify the signal they are trying to interpret!

1. size: "This resulted in particle-size distributions consisting of 92 logarithmic size classes ranging from 0.375 to 2000 µm. Grain-size statistics were calculated geometrically using the graphical method of Folk and Ward (1957) using GRADISTAT (Blott and Pye, 2001)." It sounds like you are assuming that the size of the particles you are measuring in the sediment trap is the size of the particles in the atmosphere? There is a lot that goes

into that kind of set of assumptions, so please spend at least a paragraph in the methods describing why you think this will work, previous papers which showed a relationship (or not) and what kind of assumptions it requires. Wetting an atmospheric aerosol during deposition, could either make particles coagulate or break the bonds of particles. On the other hand, material can coagulate onto the particles in the ocean, and change or process them. Already your evidence that the sizes are different at the trap and deeper down suggests changes in size or processing (or advection: see next point). Please be explicit about the assumptions you are making and justify them here in the results, and then in summary and conclusions discuss the implications of your assumptions for your work. Are these aerodynamic or geometric measurements, as a basis? Size of aerosols is tricky to measure (e.g. Reid et al., 2003) and different measurement methods get quite different results: how do your methods compare? Is there any way you can use previous measurements of size in aerosols (e.g. Reid et al., 2003; Skonieczny et al. (2013)) to help you with this problem? How do you know that these particles are from eolian deposition and not some other process?

First, the data described here are measurements of actual dust particles, and not a proxy for these dust particles. The cited paper by Bory et al. (2002) uses a proxy (Al) for lithogenic particles, but in our study we isolated the lithogenic fraction using chemicals, and performed grain-size analyses on this fraction only. As an example, Stuut et al. (2005) demonstrated the similarity between aerosol samples of sediment traps and dust found in sediment traps and in seafloor sediments. In this paper, we discuss data of deposited dust, and not of transported dust. The transportation of dust is, however, used for interpretation of the data. Seasonal changes observed in the traps are not only due to changes occurring in the atmosphere, but also in the sources of the dust: summer months are characterized by more convection due to bigger temperature differences, resulting in the uplifting of coarser particles. Also the different dust-transporting winds during the different seasons (the trade winds and Saharan Air Layer – SAL) that blow in different directions, at different altitudes and with different wind speeds cause the seasonal differences in particle size of transported dust.

We added the following lines to the revised manuscript:

*Page 16, lines 3-5:*

"In addition, increased convection in the source areas in summer, related to larger differences in temperature, can result in the uplift of coarser dust particles (Heinold et al., 2013)."

The assumption is made that the lithogenic particles we find in the sediment traps are of aeolian origin, more specifically dust originating from Africa. The sediment traps are far from the continental shelf, so riverine input of sediments can be excluded. Limited influence of major rivers is also visible when looking at (satellite) data of chlorophyll A or salinity, available from Giovanni of NASA GES DISC (http://giovanni.sci.gsfc.nasa.gov/giovanni/). Also, the lower sediment traps are positioned 880-1300 m above the seafloor (M2-M5), so resuspension of bottom sediments will not affect the sediment trap samples. When considering the large amounts of Saharan dust being transported every year, about 182 million tons (Yu et al., 2015), it is fair to consider the lithogenic fraction in the sediment traps to be of aeolian origin.

We added the following lines to the revised manuscript:

Page 5, lines 9-17:

"In this paper we argue that the lithogenic particles found in the sediment traps are of aeolian origin. The sediment traps are located far from the continental shelf, so riverine input of sediments is not affecting the samples. Limited influence of major rivers is also visible when looking at (satellite) data of chlorophyll or salinity (not shown; available form Giovanni NASA GES DISC: http://giovanni.sci.gsfc.nasa.gov/giovanni/). In addition, the lower sediment traps are positioned 880-1300 m above the seafloor, so resuspension of bottom sediments will not affect the sediment trap samples. When considering the large amounts of Saharan dust being transported across the Atlantic Ocean every year, about 182 Tg (Yu et al., 2015), any other external input is

assumed to be negligible. Stuut et al. (2005) also demonstrated the similarity between aerosol samples of Saharan dust collected off west Africa and the lithogenic fraction in sediment traps and seafloor sediments."

During the grain-size analysis, individual particles are analyzed. First, the lithogenic fraction of the sediment-trap samples (or sea-floor sediments) is isolated, and the aggregates destroyed by the three-step pretreatment procedure of boiling with chemicals to remove all organic-produced components. So any post-transport or post-depositional aggregation is also removed. It is possible that dust particles are transported as part of aggregates in the atmosphere, and also these aggregates are destroyed during the sample preparations, resulting in the measurement of smaller particles. Additional processing of the particles in the ocean would not result in larger particles, only possibly in smaller particles. But since the processing before analysis is identical for every sample, the results can be directly compared to each other. In total, the particle-size distribution that is measured is at the very least an underestimation of the actual size of 'particles' (including aggregates) that are being transported. Grain-size analysis of 1-year accumulative samples from floating dust-collectors at our transect (unpublished results) have shown to be almost identical to the underlying sediment-trap samples, confirming again that the dust we find in the sediment traps is representative of the atmospheric dust.

Difference in particle size between the upper and lower traps could be the result of advection, but this is thought to be minimal. More likely is the larger catchment area for the deeper traps, resulting in the sampling of slightly more particles and particles of different sizes. A new figure added to the revised manuscript (Fig. 2) illustrates the similarity between the upper and lower traps at station M4, see also the next comment.

The measurements described here are geometric measurements, and were performed with a Coulter laser diffraction particle sizer, as described in the Material and Methods section of the manuscript. These measurements are the same as Skonieczny et al. (2013) performed, using a Malvern Mastersizer 2000 laser diffractometer. So these data can be compared almost directly (Konert and Vandenberghe, 1997). The geometric method may pose a potential problem for 'lightweight' mica particles: due to their platy shape they have a smaller aerodynamic size than geometric size, and as described in the manuscript [in Introduction and Discussion] as well as in Stuut et al. (2005), this is expressed as a secondary mode in the grain-size distribution. Therefore, it was decided to describe the grain-size data with the modal value of the distribution, which accounts less for these coarser particles with smaller aerodynamic size.

2. advection and sedimentation rates: The second issue is the relationship of what enters the top of the ocean and what is deposited in the sediment traps. Previous studies (Bory et al., 2002) have suggested that productivity could modulate the transfer rate between the top of the ocean, and the cores. We know that the dust has to be carried with the current as it floats downward: how far downward? (Han et al., 2008; Siegel and Deuser, 1997) show that it really can be quite far. If it also seasonally being modulated, that would really mess up your signal!

Since we have recovered both sediment traps, the upper (1200 m) and lower (3500 m), at two of the stations (M2 and M4), we can compare the data between these two traps. Although sediment fluxes are not the scope of this paper, a new Figure has been added to the revised manuscript (Fig. 2), showing photos of the sediment-trap bottles after recovery, with high levels of sediments in two samples (sample 12 and 24, collected during spring and fall, respectively). These high fluxes are present in both the upper and the lower traps, in the same sampling cup. This demonstrates the similarity in sediments received for both sediment traps, and that lateral advection is minimal. Since the sampling interval is only 16 days, it means that the downward transport velocity of these sediments is at least 140 m day$^{-1}$. However, it seems that the high flux in sample 12 of the upper trap is distributed over sample 12 and 13 in the lower trap. This demonstrates that there is a small time-lag between the two traps, of no more than a few days, due to the time it takes for the particles to settle. This could also be true for sample 24, however there is no sample directly after this last sample.

We added the following lines to the manuscript, in addition to Figure 2:

*Page 4, lines 8-10 and page 5, lines 1-8:*

"Since both the upper and lower traps are recovered for two of the five stations (M2 and M4), this allows for a direct comparison between the two depths. The upper and lower sediment traps are in very good accordance with

each other, demonstrated by images of the sediment-trap bottles after recovery (Fig. 2). Two samples, sample 12 and 24, have a much higher flux than the other samples, and these high-flux samples are present in both the upper and lower trap. Since the sampling interval is only 16 days, it means that the downward transport velocity of the sediments between the traps is at least 140 m day$^{-1}$ and most likely much higher. It also shows that the sediments are deposited in a vertical way down to both sediment traps. It seems however that the higher flux observed in sample 12 of the upper trap is distributed over sample 12 and 13 of the lower trap. This demonstrates that there is a small time-lag between the two traps, of no more than a few days, due to the time it takes for the particles to settle. This could also be true for sample 24, however there is no sample directly after the last sample of the sediment trap."

3. deposition rates. What are the deposition rates you are getting? Are they consistent with your assumptions? Are the deposition rates reasonable? Are they the same in the sediment traps as the sediment below? Please describe this a bit more.
Sediment fluxes of the sediment traps are beyond the scope of this paper, and a paper showing these results is in preparation at the moment. What we can conclude from the two events that were registered in the same bottles of traps that are positioned at a vertical distance of more than 2 km, is that settling rates through the water column are at least 140 m day$^{-1}$ and probably higher. In any case they are in the order of days, not seasons. For the seafloor sediments, however, it is difficult to give a sedimentation rate for the upper 1 cm, since it is difficult to date these samples. Given the fact that they could be hundreds to thousands of years old, it is to be expected that they cannot be compared with our present-day samples directly. Mulitza et al. (2010) have shown that since the arrival of Portuguese colonists in Africa about 300 years before today, the change in land-use changed the emissions of northwest African dust dramatically. As the seafloor sediments presented in our manuscript could easily be older than 300 years, we present this line of reasoning as an explanation for the observed difference.

More details
"However, grain sizes in the seafloor sediments are substantially finer than found in the sediment-trap samples, and the downwind decrease in grain size is also less steep for the seafloor sediments." What does this mean for interpretation? The more processed the cores, the finer they look? Or that they are being dissolved? Or that they are advected from farther upstream? I find this observation very difficult to understand, and makes me doubt your methodology.
As stated above, the main difference between the sediment traps and the seafloor sediments is the timing of the samples: the sediment traps have a very fixed time-resolution of 16 days. The seafloor sediments, however, are the result of accumulation of hundreds of years in the top centimeter alone. The grain-size distribution for the seafloor sediments is therefore an average of all these years, and the fact that the particle size is smaller means that over this long time period the dust was finer-grained than it is today (as found in the sediment traps). Dissolution of the particles is not likely, since most particles are quartz particles which are very resistant to this kind of processing.

"Since the seafloor sediments represent a longer time average of Saharan dust deposition than the sediment-trap samples, it implies that the downwind fining is a long-lived trend." How long is the time average for the sedments on the seafloor compared to the traps?
See the comments above. The seafloor sediments are an accumulation of hundreds of years of sediments, and it is difficult to date the top centimeter. Typical accumulation rates for deep-sea sediments, however, are 1-5 cm kyr$^{-1}$ (Anderson, 2007), in this area possibly even lower, indicating that the top centimeter alone represents several hundreds to thousands of years.

"Mahowald et al. (2014) hypothesize that dust in the high atmosphere is finer grained than in the lower atmosphere, which is in turn finer than the deposited dust, due to the preferential settling of coarse particles. However, we observed giant particles (≥100 m) as far as station M4 (49° W; approximately 3500 km from African coast) (Fig. 4)." On the surface of it, these two statements have nothing to do with each other, since one is talking about vertical height in the atmosphere and the other is talking about horizontal distance from

Africa. You seem to be implying that they are somehow contradicting each other, but it doesn't seem possible to infer from distance downwind anything about vertical structure tof the atmosphere?.

These two statements were meant to illustrate the preferential settling of coarse particles, however still giant particles are observed at great distances from the source. We removed this sentence in the revised version of the manuscript, since it is also stated in the previous sentence, and rephrased the two paragraphs:

*Page 13, lines 5-17, and Page 14, lines 1-4:*

"The grain size of dust decreases with increased distance from the source (Glaccum and Prospero, 1980;Goudie and Middleton, 2006;Mahowald et al., 2014;Stuut et al., 2005): coarse particles have a higher settling velocity and smaller particles can be transported over greater distances (Gillette, 1979;Tsoar and Pye, 1987). This mechanism accounts for the downwind fining observed in both the sediment traps and the seafloor sediments along the trans-Atlantic transect (Fig. 3). However, we observed giant particles ($\geq$100 µm) at station M3 (38˚ W; approximately 2400 km from the African coast) (Fig. 5), and also mica particles, whose platy shape allows for aerial transportation over greater distances (Stuut et al., 2005). Such coarse particles are generally not incorporated into climate models (Kok, 2011). Only a handful of these coarse particles are found in the samples, however when considering these are 1/25 splits of the original samples, collecting sediments over only 1 m$^2$ of ocean, over a time period of only 16 days, this means that the amount of giant particles being transported over the Atlantic Ocean is substantial. This underestimation of the coarse size fraction may have its origin in the sampling of dust of specific size classes, e.g. $PM_{10}$ and $PM_{2.5}$, which form the basis of the guidelines from the World Health Organization (WHO, 2006) on fine-grained particles.

Since the seafloor sediments represent a longer time average of Saharan dust deposition than the sediment-trap samples, it implies that the downwind fining is a long-lived trend. However, the modal particle size of the sediment-trap samples is substantially coarser than that of the seafloor sediments at the same stations along the transect. […]"

"The particle-size distribution found in the sediment-trap samples closely resemble Saharan dust sampled directly from the atmosphere, which has modal grain sizes varying between 8 and 42 µm (Stuut et al., 2005)."
This is really important, but you don't say where this observation is made? Size is varying along the transect in the atmosphere also: : :.. what type of observation is this? What kind of uncertainties are in that method (i.e. look at (Reid et al., 2003))

For more information on the scientific expedition during which these samples were collected and their exact sampling locations we refer to Figure 1 from Stuut et al. (2005).

We modified the following lines in the revised manuscript:

*Page 14, lines 4-7:*

"The particle-size distributions found in the sediment-trap samples closely resemble Saharan dust sampled directly from the atmosphere, by shipboard dust samplers along a transect off the West African coast, which has modal grain sizes varying between 8 and 42 µm Stuut et al., 2005). This is in close resemblance with the observed modal grain size of 4 – 32 µm in the sediment traps."

"By contrast, modal grain sizes in the underlying seafloor sediments range between 4 and 6 µm. Since the seafloor sediments represent a longer time period, this suggests that Saharan dust was significantly finer in the recent past than it is today." And in the conclusions: "Coarser dust found in the sediment traps opposed to the

seafloor sediments could result from emission of coarser dust due to the onset of commercial agriculture in the 19th century." This is a huge jump, which seems incredibly unlikely. Most likely there is ocean processing: : :.

The proposed mechanism is one of many possible causes for an increase in particle size over the past few hundred years. It is not meant to be conclusive or the sole mechanism behind this change in particle size over the last few centuries.

We modified the following lines in the revised manuscript:

*Page 14, lines 7-14*:

[revised manuscript text omitted]

"We have shown seasonal and spatial changes in Saharan mineral dust transport and deposition across the Atlantic Ocean by means of sediment-trap sampling between October 2012 and November 2013, and seafloor sediments at the same stations." So at this point, this statement has not been proven: you have only shown sea floor sediment changes in size. It is interesting that you see these trends, but anything about the atmospheric aerosols is speculation.

Indeed no seasonality can be seen for the seafloor sediments, unlike the sediment traps that sample at very high resolution (16 days).

We rephrased the following lines in the revised manuscript:

*Page 19, lines 3-7:*

"We have shown seasonal and spatial changes in Saharan mineral dust transport and deposition across the Atlantic Ocean by means of sediment-trap sampling between October 2012 and November 2013, and spatial changes in the seafloor sediments at the same stations. Our results show strong seasonal variations and significant fining in particle size with increasing distance from the source in the sediment trap samples, with modal particle diameters ranging from 4 to 32 µm."

Han, Q., Moore, J.K., Zender, C., Measures, C., Hydes, D., 2008. Constraining oceanic dust depostion using surface ocean dissolved Al. Global Biogeochemical Cycles 22, doi:10.1029/2007GB002975.
Reid, J.S., Jonson, H., Maring, H., Smirnov, A., Savoie, D., Cliff, S., Reid, E., Livingston, J., Meier, M., Dubovik, O., Tsay, S.-C., 2003. Comparison of size and morphological measurements of dust particles from Africa. Journal of Geophysical Research 108, 8593: doi:1029/2002JD002485.
Siegel, D.A., Deuser, W.G., 1997. Trajectories of sinking particles in the Sargasso Sea: modeling of statistical funnels above deep-ocean sediment traps. Deep-Sea Research 44, 1519-1541.

**Anonymous Referee #2** (Referee comments in black, our reply and changes in the manuscript in green)

Van der Does and colleagues present their preliminary findings for one year of data from a multi-year sampling campaign, aimed at retrieving samples of dust deposited to the equatorial Atlantic Ocean (at latitudes _12 N), utilizing a transect of moored sediments traps at different water depths, downwind from the North African dust sources. The aim of this ambitious project to better constrain the evolution of the North African dust plume is of certain interest, and it is positive for the atmospheric and dust communities to be informed about these preliminary findings. This is a very interesting study, and the manuscript is in general well organized and quite clear. Nonetheless I do revise three major aspects that would deserve some revision, in order to clarify the work and perhaps improve the possible interpretation of the data.

General major comments

The particle size distributions are central in the manuscript. Nonetheless the descriptive metrics that are used for some of the diagnostic plots are only briefly mentioned. I think it would be very important to clearly show a validation of the metric used (e.g. mode of distributions fitted using GRADISTAT) against the specific observational data, before following with the discussion. This is particularly relevant since many of the samples show an apparent bi-modal distribution.

In this case we chose to represent the grain-size distributions with the mode instead of the median grain size, since the mode shows the most-occurring value. The median would also account for any secondary modes. However, these secondary modes, as also described in the manuscript, most likely represent platy mica particles, that due to their shape have a larger geometric size than their aerodynamic size. Therefore, we chose the mode to represent the grain-size distributions of our samples to better represent the geometric grain sizes.

In the discussion of the data, the coarseness of grain size distributions is assumed to mimic the behavior of dust deposition flux, despite the fact that such information is not reported. In addition, a full comparison among different samples is hampered by the lack of this piece of information.

In the discussion we speak only of increased dust emission (as seen from AOD and satellite data) in line with an increase in particle size of the dust found in the sediment traps which could be related. Specific flux data for the sediment traps should give more insight in this statement, but that is beyond the scope of this paper.

Some of the interpretations of the data provided in the discussion remain rather speculative. Please see the specific comments below.

Specific comments

2 / 8-9. Which is the mechanism that would explain this statement, in relation to the previous sentence?

As discussed in the discussion section of the paper, increased wind velocities in the SAL, combined with the higher elevation of the dust particles, results in the transportation of coarse particles over greater distances. Also convection within the SAL keeps coarse particles suspended. When high up in the atmosphere it takes more time for particles to settle down, and when also transported in a lateral sense these particles can reach further across the Atlantic Ocean.

3 / 26-32. I would argue that the point here is that we need to quantify how far a significant number of those particles can travel, to both (a) constrain the inputs to the ocean and (b) be able to estimate if, because of their actual amount, they are in fact relevant in terms of direct radiative effects or rather they could actually be ignored from this point of view, as typically done so far in models. That is why it would be important to have dust deposition fluxes associated to the size distribution data. If this piece of information is not available, the discussion should take into account this fact, and the interpretation of similarities / differences in the samples should be pondered accordingly.

Dust fluxes of the sediment traps are beyond the scope of this paper. With this information, however, it would be difficult to give an accurate number of the amount of particles larger than a certain diameter. The grain-size

distributions only show relative volumes or the particles, and not actual particle counts. However, given that about 182 Tg of dust are transported over the Atlantic Ocean every year (Yu et al., 2015), the samples collected for our study have a temporal resolution of only 16 days, the collection area of the sediment traps is 1 m$^2$, the analyzed split of these samples is 1/25, and that we can see at least a handful of these giant particles per sample illustrates that the amount of these coarse particles transported in the atmosphere must be substantial.

4 / 30. Briefly, why did you discard some of the traps?

See caption of Figure 1; three of the ten traps could not be recovered due to material failure and rough weather during recovery.

We modified the following lines in the manuscript:

*Page 3, lines 37-39, and page 4, lines 1-2:*

"This paper presents the results of successful sampling by seven sediment traps on the five moorings from 19 October 2012 to 7 November 2013 (Stuut et al., 2013). These include three of the upper (1200 m) sediment traps located at mooring stations M1, M2 and M4, and four lower (3500 m) sediment traps at stations M2, M3, M4 and M5 (Fig. 1, Table 1). Three of the ten sediment traps could not be recovered."

5 / 19-20. Please clarify if you refer to radius or diameter, here and throughout the manuscript.

All particle sizes referred to in the manuscript are described as equivalent-sphere diameter of the particle.

We modified the following lines in the manuscript:

*Page 6, lines 6-8:*

"This resulted in particle-size distributions consisting of 92 size classes ranging from 0.375 to 2000 µm describing the equivalent-sphere diameter of the particle. Modal particle size is also expressed as particle diameter."

5 / 20-21. As indicated above, please discuss much more extensively this aspect. For instance, describe how the method works, and show the comparison of the full distribution and the metric (mode) for two-three representative cases, e.g. a typical sample for each Winter, Summer, Spring from Figure 5. This should highlight how the metric vary according to the distribution's shape, thus help better understanding/constraining the following interpretations. This should also highlight whether the choice of the metric is the best option or better ones could be adopted in this case.

For a discussion why the mode should be used for describing the particle size of the dust, please refer to the first comment (page 8 of this letter). A note on why the method of Folk and Ward (1957) was used: initially we chose the median diameter to describe the particle size, and in this case there is a difference in what the Coulter Laser Diffraction Particle Sizer (LS13 320) calculates. This method also allows for better comparison to other methods (Blott and Pye, 2001). For the modal diameter, however, the GRADISTAT program recalculates the particle-size classes, resulting in different modal diameters than based on the size classes of the Coulter. We decided to change these modal diameter to the original mode given by the Coulter (the raw data), and not refer to the Folk and Ward method. We changed all the graphs accordingly. This does not change the trends we see in particle size, but alters the absolute values of the modal diameter slightly.

We modified the following lines in the manuscript:

*Page 6, lines 6-8:*

"This resulted in particle-size distributions consisting of 92 logarithmic size classes ranging from 0.375 to 2000 µm describing the equivalent-sphere diameter of the particle. Modal particle size is also expressed as particle diameter."

This also resulted in small changes regarding the modal particle size shown in Figures 2, 6, 7, 8 and 11 (Figures 3, 7, 8, 9 and 12 in revised manuscript).

5 / 30. Did you only simulate four days? Later in the manuscript (9 / 31-32) it sounds like you may have selected four days out of a larger ensemble? Is that the case? If so, it would be interesting to see those. If not, how did you exactly determine that those would be representative days?

For the backward trajectories, only four days were simulated. This is a balance between a good insight of where the starting point of the air parcel is, and modelling uncertainties with simulating longer time periods. It is also used more frequently in literature (e.g. Stuut et al. (2005)). Only a basic overview was made for backward trajectories starting at station M1 for the entire sampling period, to see if there would be seasonal differences. This resulted in a very wide range of trajectories, showing no clear seasonal trend, and which can't be visualized in a clear way. The cases shown in the manuscript were chosen as a clear example for summer and winter dust transportation.

6 / 1. Do you have dust deposition flux data? I believe that all the comparisons among the samples in this study and the derived interpretations are subject to the limitation of not being associated to dust deposition fluxes. Therefore only partial information is available to derive conclusions.

Dust fluxes of the sediment traps are beyond the scope of this paper. We realise that they are of large scientific interest for many different scientific disciplines. Therefore, they are presently being determined, however not for this particular publication.

7 / 14-17. As already mentioned, I think that the point is not whether a handful of giant particles make it a great distance, but rather how many and how far. If they appear to be quantitatively important, then this suggest that models should account for that, and they will need data to constrain their results. Hopefully your study will help addressing this issue!

As already mentioned in one of the previous comments (on page 8-9 of this letter), it is difficult to give an accurate number of the amount of particles larger than a certain diameter. However, given that about 182 Tg of dust are transported over the Atlantic Ocean every year (Yu et al., 2015), the samples collected for our study have a temporal resolution of only 16 days, the collection area of the sediment traps is 1 $m^2$, the analyzed split of these samples is 1/25, and that we can see at least a handful of these giant particles per sample illustrates that the amount of these coarse particles transported in the atmosphere must be substantial.

7 / 17. "Preferentially" vs what? Please clarify this sentence.

The platy particles are more easily transported over greater distances than deposited close to the source, opposed to larger spherical particles. We acknowledge this sentence is not clear, and therefore removed it from the revised manuscript. We left the following statement in:

Page 13, lines 9-11:

"However, we observed giant particles (≥100 µm) at station M3 (38˚ W; approximately 2400 km from the African coast) (Fig. 5), and also mica particles, whose platy shape allows for aerial transportation over greater distances (Stuut et al., 2005)."

7 / 25-30. This paragraph seems very speculative: there is no support to it in the discussion, and no time control is reported about the age of those seefloor sediments.

The proposed mechanism is one of many possible causes for an increase in particle size over the past few hundred years. It is not meant to be conclusive or the sole mechanism behind this change in particle size over the last few centuries.

We modified the following lines in the revised manuscript:

*Page 14, lines 8-14*:

"Since the seafloor sediments represent a longer time period, this suggests that Saharan dust was significantly finer in the recent past and increased over the last centuries. Deposition of coarser dust is in line with increased emission as a result of human activity since the nineteenth century due to commercial agriculture (Mulitza et al., 2010). Not only does increased human activity in the source region increase dust emissions, it also enables larger particles to be emitted (McTainsh et al., 1997), which could cause the particle size of the deposited Saharan dust to become gradually coarser over time, as we see now in the sediment traps."

8 / 11-15. Here you seem to suggest a direct relation between coarse grain size and high dust load (or AOD), and for extension to a high deposition flux? The reported study of Skonieczny et al. (2013) on the other hand shows coarser dust deposition at M'Bour, Senegal, associated with the season of low dust deposition flux. How would you justify your assumption in light of that? I think that absolute magnitudes of size distributions could help here in two different ways, most importantly with reference to Figures 6, 7, 8, 11. First, absolute values of particles concentrations (i.e. counting statistics on the direct output from the particle counter) may help to understand if the "shoulders" associated to the larger particles are actually statistically significant in all cases. One can see "tail effects" associated to sometimes individual large particles in low concentration samples such as from ice cores (e.g. Albani et al., 2012). This piece of information should be considered together with the choice of the mode as a metric to compare those samples. Second, even when samples are screened against possibly noisy signals, any interpretation on the actual quantitative transport potential (whether with season, or distance, or depth) of giant particles remains speculative without deposition flux data. The same way, in order to trace the spatial evolution of the North African dust plume, size distributions are necessary but not sufficient. Comparing sediment records from the Atlantic on different size ranges in fact yield surprising results, demonstrating the importance of considering both size distributions and fluxes (Albani et al., 2015). If this piece of information is missing, then the discussion should be extended to discuss the possible limitations of the derived interpretations.

In the discussion we speak only of increased dust emission (as seen from AOD and satellite data) in line with an increase in particle size of the dust found in the sediment traps which could be related. The study by Skonieczny et al. (2013) shows a similar trend as what we find in our study, with coarser dust deposition during summer. The fact that we see increased dust transportation (as reflected by the AOD) does not have to correspond with increased dust deposition. Dust fluxes for the sediment trap samples will give more insight in these mechanisms, but they are beyond the scope of this paper. However, these data will not provide number distributions of the particle size, as the method for particle size analysis describes the results as relative volumes of the particles, and not actual particle counts.
Our results are different from dust collected in ice cores, since the sediment trap samples have much more dust in them. For the ice core samples, these "tail effects" are much more significant since there is much less dust in each sample. In addition, the Antarctic ice sheet is 2315 m above sea level, decreasing the probability of coarse particles being uplifted to these heights and deposited on the ice sheets.

8 / 24-28. Interesting approach!
Thank you!

8 / 31-32. Quite the opposite. I cite: "On balance, the measurements (Fig. 4) indicate that dust PSD is independent of the wind speed at emission. This conclusion is supported ..."

Cited from Mahowald et al. (2014): (p. 64) "The results suggest that an increase in wind speed can be associated with a small (0.15 µm) increase in dust particle size downwind of the sources (Fig. 11). This is consistent with the paleoclimate interpretation that stronger winds will carry larger particles. "

And (p. 67): "The size of individual particles is to a large extent set at emission, [...]"

We modified the following lines in the revised manuscript:

*Page 15, line 16 and page 16, lines 1-5*:

"Mahowald et al. (2014) argue that the dust particle size does not depend on wind speeds at emission. However, high wind velocities in the SAL of $>7$ ms$^{-1}$ (Tsamalis et al., 2013) enables coarser dust particles to remain in suspension in summer, and due to the high altitude these coarse particles are transported over great distances. In addition, increased convection in the source areas in summer, related to larger differences in temperature, can result in the uplift of coarser dust particles (Heinold et al., 2013)."

9 / 3. I would suggest changing "these air layers" with something like "the starting points for back-trajectories calculations".

We agree. We modified the following lines in the revised manuscript:

*Page 16, lines 8-11*:

"The altitudes of the starting points of these backward trajectories were chosen in accordance with the hypothesized heights of the dust-carrying air layers, as demonstrated in Figures 11A and -B, with the lowest (500 m) elevation representing winter dust transport and the highest (3500 m) elevation representing summer dust."

9 / 8-9. This sentence is not very clear, please rephrase.

We agree. We modified the following lines in the revised manuscript:

*Page 16, lines 11-22*:

"In winter (Fig. 11C), the higher trajectory is not originating from the African continent, and therefore the winds at these altitude are unlikely to transport dust to the sample location. The lower trajectory has a more eastern origin, and air layers at this altitude could be transporting dust (Fig. 11A), picked up from the surface and brought to higher altitudes. By contrast, in summer (Fig. 11D) this situation is reversed: the higher trajectory has a more continental origin and is the most likely dust-carrying air layer over the lower trajectory. The elevation profile shows that this high-elevation trajectory started at lower altitudes, but upon reaching the coastline it was uplifted to about 3500 m AGL (Fig. 11D, bottom panel). This is in accordance with how the Saharan Air Layer (SAL) is described, when dust-carrying air from the continent is uplifted by a cool marine inversion layer (Carlson and Prospero, 1972;Prospero and Carlson, 1972). This inverted air layer is visible in the 500 m air layer, moving in an opposite direction, from west to east. After this sharp increase in altitude, the trajectory decreases in altitude, which persists across the Atlantic Ocean (Tsamalis et al., 2013)."

9 / 12-14. It seems that here "air-layer" is used to indicate "air parcel trajectory"?

This comment is dealt with by the change in the manuscript discussed above.

9 / 15-18. You are not showing this. Please at least provide some reference.

We modified the following lines in the revised manuscript:

*Page 17, lines 8-10*:

"The summer season is also characterized by an increased number of more intense dust storms (e.g. Adams et al. (2012)). From May to September, dust is almost continuously emitted from the African continent, as shown by satellite images (MODIS Terra and Aqua satellites; NASA Worldview)."

9 / 18. "Increased deposition": where?

What is meant here is "increased deposition of coarse particles", relative to the other particles, so coarser dust deposition.

9 / 21-22. How do you know? You do not show any information about the atmospheric column above.

We can only note the apparent coincidence of increased particle size and increased precipitation. It can be one of many mechanisms related to deposition of coarser particles.

We modified the following lines in the revised manuscript:

*Page 17, lines 12-16*:

"Increased deposition of coarse particles can also be caused by increased precipitation in summer and fall, as opposed to almost no precipitation in winter and spring (Fig. 12). This was also noted off northwest Africa related to wet deposition was also noted by Friese et al. (In press). Increased precipitation at station M1 seems to coincide with increased modal grain sizes, and this relation commences with lowest precipitation early June 2013."

9 / 23-24. Again, it is not clear whether the mode is a good metric to compare bi-modal distributions.

It describes the trends that are most clear in seasonality of the dust particle size. The main mode that occurs in all samples is thought to be made up of quartz particles, the second mode consists of mostly coarse, platy mica particles, and Fig. 12 is used to better demonstrate this second coarser mode (see also previous comments). Most samples however do not show a bi-modal grain-size distribution, hence the choice for comparing the mode of the different samples.

9 / 25-30. How does a laser particle counter sees a flat particle? Overestimate it's spherical equivalent diameter? See e.g. Reid et al. (2003). How do you interpret this in your data, and according to the evolution of size distribution with distance from the source?

The laser particle sizer measures the diameter of the particle as it is oriented towards the laser beam. As the sample is being constantly homogenized by a magnetic stirrer, there is no preferred orientation for particles and thus random. As a result, the flat particles will be measured (in theory) in an infinite number of ways, and hence detected as a smaller or larger particle, depending on its orientation. The result would be an average of the smallest and largest diameter of equivalent spherical particle, but since it is described as volume percentage, the larger would have more influence on the grain-size distributions. Therefore it was chosen to describe the data with the modal diameter, and separately address the second mode of the distributions (see also previous comments).

9 / 30-32. As already mentioned, if more back-trajectories calculations were performed, it would be interesting to see them.

See answer to a previous comment (page 10 of this letter).

10 / 1-6. Also in this respect, absolute values of concentration and most importantly dust deposition fluxes might shed some light on the issue. In addition, a little more discussion on the fate of particles throughout the water column and the expected relation to the corresponding surface water and atmosphere could be added here.

Since we have recovered both sediment traps, the upper (1200 m) and lower (3500 m), at two of the stations (M2 and M4), we can compare the data between these two traps. A very clear example is visible for the samples at M4, where two samples (sample 12 and 24, collected during spring and fall, respectively) with very high fluxes are present at both the upper and the lower trap, in the same sampling cup. Since the sampling interval is only 16 days, it means that the downward transport velocity of these sediments is at least 140 m day$^{-1}$. Sediment fluxes are beyond the scope of this paper, however a new Figure has been added to the present manuscript showing sediment-trap bottles after recovery, with high levels of sediments in the aforementioned samples (Fig. 2 of revised manuscript). This demonstrates the similarity in sediments received for both sediment traps, and that lateral advection is minimal.

10 / 11. Please add also here in the conclusions whether you refer to particle radius or diameter.

We modified the following lines in the revised manuscript:

*Page 19, lines 3-7*:

"We have shown changes in Saharan mineral dust transport and deposition across the Atlantic Ocean by means of sediment-trap sampling between October 2012 and November 2013, and seafloor sediments at the same stations. Our results show strong seasonal variations and significant fining in particle size with increasing distance from the source in the sediment trap samples, with modal particle diameters ranging from 4 to 32 µm."

10 / 11-12. As indicated earlier, this statement is so far very speculative.

See also previous comments on page 10-11 of this letter. The proposed mechanism is one of many possible causes for an increase in particle size over the past few hundred years. It is not meant to be conclusive or the sole mechanism behind this change in particle size over the last few centuries.

We modified the following lines in the revised manuscript:

*Page 19, lines 7-9*:

"Coarser dust is found in the sediment traps opposed to the seafloor sediments, in line with increased emission and coarser dust due to the onset of commercial agriculture in the 19[th] century."

10 / 22-23. From your study, one would expect to learn how many.

See also previous comments on page 8-9 and 10.
Dust fluxes of the sediment traps are beyond the scope of this paper, and it is difficult to give an accurate number of the amount of particles larger than a certain diameter. However, given that about 182 Tg of dust are transported over the Atlantic Ocean every year (Yu et al., 2015), the samples collected for our study have a temporal resolution of only 16 days, the collection area of the sediment traps is 1 m$^2$, the analyzed split of these samples is 1/25, and that we can see at least a handful of these giant particles per sample illustrates that the amount of these coarse particles transported in the atmosphere must be substantial.

Figure 2. Please differentiate the markers based on the depth for M2 and M4.

We modified the indicated figure in the revised manuscript (Figure 3 in revised manuscript) and the figure caption.

Figure 7. Could you provide a brief explanation about those outliers?

The measurements for these two samples are out of the entire range of particle size, for all stations, and seem very unlikely. This may be related to analytical or processing errors, since there are many steps involved before the particle size is analyzed. Out of 168 samples analyzed for this paper, only two unrealistic outliers appear, which we consider reasonable. For this reason, we chose to not elaborate on this in the manuscript.

"Coarser dust is found in the sediment traps opposed to the seafloor sediments, in line with increased emission and coarser dust due to the onset of commercial agriculture in the 19$^{th}$ century."

Although it is generally understood that the SAL is transported westward over the Atlantic, the authors draw many conclusions of the seasonal altitude dependence of air mass transport and at only one trap location (M1). What would strengthen the argument regarding the impact of transport conditions and seasonal climate patterns on particle deposition/size is an ensemble or cluster analysis of HYSPLIT trajectories. The authors do state, "However, backward trajectories calculated over the entire sampling period do not suggest this: : :" which indicates that more trajectories were simulated. It would be helpful to show these to clearly show the seasonal variability. It would also be useful to conduct HYSPLIT analyses at all of the trap locations to better connect the sites and perhaps show that transport over the trap farthest from Africa does not experience as much transport as the trap closest.

More focus is on sampling station M1 since this is closest to the source, and the differences between seasons are greatest here. However as can be seen from the grain-size data, seasonality is present for all the five stations.

Only a basic overview was made for backward trajectories starting at station M1 for the entire sampling period, to see if there would be seasonal differences. This resulted in a very wide range of trajectories, showing no clear seasonal trend, and which can't be visualized in a clear way. The two cases shown in the manuscript were chosen as a clear example for summer and winter dust transportation. Again station M1 was chosen for this, as it is located closest to the source. Backward trajectories for the other stations could be useful, but since the distance to the source is greater there are more uncertainties, and it will be a futile task to attempt to illustrate air-layer trajectories for the entire sampling period for all stations. From satellite images it becomes very clear that dust is transported from the African continent over the Atlantic Ocean and the sampling stations, and the backward trajectories were intended to show typical summer and winter transport of dust, and to illustrate the seasonal differences.

General comments

The figures present data from a number of sources (i.e., MODIS and particle imaging). Although the captions to these figures briefly describe these data sources, they should be more comprehensively described in the methods section. As an example, what instrument was used to image the particles? How many images were acquired? Was this conducted for all samples? With respect to MODIS, provide at the very least a brief description of the satellite and how the data were acquired. For the precipitation, was this acquired from TRMM? Over what domain?

The microscope images were performed with a normal light microscope, and the ones shown were chosen to act as an example for the coarse particles found in the samples. This was not done for all 168 samples, however from the grain-size distributions it is clear that these coarse particles are present in most samples. The image acts as an aid to illustrate that these coarse particles are also solid quartz particles, and not only platy mica particles, and that the coarse particles measured are not simply aggregates of smaller particles.

In the revised manuscript, we added more microscope images of large particles (*Fig. 5*), and modified the figure caption:

"Figure 5. Light-microscope images of large dust particles from the lower (3500 m) traps at station M2 (13° N, 37° W; A and B) and at station M3 (12° N, 38° W; C and D). Both stations are situated at more than 2000 km from the African source. A: Large quartz particle (diameter approximately 180 µm over long axis) from sample 1 (October 19 – November 4, 2012). B: Large quartz particle (diameter approximately 290 µm over long axis) from sample 1. C: Large quartz particle (diameter approximately 200 µm over long axis) from sample 4 (December 6 – 22, 2012). D: Large mica particle (diameter approximately 86 µm over long axis) from sample 4."

We also added more information about the MODIS images and precipitation data in the respective figure captions.

We added the following paragraph to the Materials and methods section of the revised manuscript:

*Page 6, lines 22-25:*

"Data for Aerosol Optical Depth (AOD) and daily precipitation were obtained from the Giovanni online data system, developed and maintained by the NASA GES DISC. AOD data was obtained from MODIS Terra, at monthly resolution and averaged over the respective seasons. Daily precipitation data from TRMM was used, averaged over the area between 11 - 13° N and 22 - 24° W (station M1)."

Specific comments

Page 2, line 19: Most people know what CALIPSO is, but do define the acronym.

We modified the following lines in the revised manuscript:

*Page 1, lines 30-33*:

"CALIPSO (Cloud-Aerosol Lidar and Infrared Pathfinder Satellite Observation) lidar measurements between 2007 and 2013 show that annually 182 Tg of African dust leaves the African continent towards the Atlantic Ocean, 132 Tg reaches 35° W, and 43 Tg reaches as far west as 75° W (Yu et al., 2015)."

Page 9, line 23: Only sand can be this size? What about large minerals? This seems like a vague definition without any measurements of the mineralogy.

In sedimentology, the term "sand" is used as a classification of particle size: any mineral particles between 63 and 2000 μm. Hence, the term sand does not imply anything about its properties; any material in this size range may be called sand.

Anderson, D. M.: Paleoceanography, in: Encyclopedia of Quaternary Science, edited by: Elias, S. A., Elsevier, 1599-1609, 2007.

Blott, S. J., and Pye, K.: GRADISTAT: A Grain-Size Distribution and Statistics Package for the Analysis of Unconsolidated Sediments, Earth Surface Processes and Landforms, 26, 1237-1248, 10.1002/esp.261, 2001.

Konert, M., and Vandenberghe, J.: Comparison of laser grain size analysis with pipette and sieve analysis: A solution for the underestimation of the clay fraction, Sedimentology, 44, 523-535, 10.1046/j.1365-3091.1997.d01-38.x, 1997.

[revised manuscript text omitted]
 lower (3500 m) traps at station M2 (13° N, 37° W; A and B) and at station M3 (12° N, 38° W; C and D). Both stations are situated at more than 2000 km from the African source. A: Large quartz particle (diameter approximately 180 μm over long axis) from sample 1 (October 19 – November 4, 2012). B: Large quartz particle (diameter approximately 290 μm over long axis) from sample 1, C: Large quartz particle (diameter approximately 200 μm over long axis) from sample 4 (December 6 – 22, 2012). D: Large mica particle (diameter approximately 86 μm over long axis) from sample 4.

**3.2 Seasonal grain-size trends**

[revised manuscript text omitted]

---

## Author Response (AR2)

Dear Dr. Schwarz and reviewers,

Thank you once again for your time on reviewing our manuscript. We carefully considered all the comments and revised the manuscript accordingly. In this document, the questions of the editor are answered in red, for reviewer 2 in green and for reviewer 3 in blue. Please find attached the revised manuscript, with tracked-changes with respect to the original manuscript.

Editor comment
Dear Dr. Van der Does and co-authors -

Thank you for your detailed response to the reviewers, and your improved manuscript. As you point out in the response, this manuscript represents a marriage across disciplines. Given that ACP is focused on the atmospheric issues addressed here, my sense is that a little bit more concession to the target community is called for.

For example, the widely raised question from the reviewers about potential biological sources of the lithogenic particles; I imagine that they were thinking (as I did) of the marine organisms that contribute their shells to limestone in the form of calcium carbonate. Could you more directly address this issue?

As we have tried to explain in the methods section of the paper, all the biogenic particles of marine origin are chemically removed: this includes all organic matter, (biogenic) carbonates and biogenic silica. What is left is the insoluble or lithogenic fraction, which is only sourced in atmospheric mineral dust. This may potentially include a small fraction of volcanogenic and cosmogenic particles as well (Plane, 2012), however on the other hand, carbonate dust particles and organic particles of non-marine origin are also removed from the samples by the chemical treatment applied. Since this is the case for every sample that we analyzed in this study, they can be compared directly to each other. In addition, we know that most of Saharan dust is made up of siliciclastics (e.g. Scheuvens and Kandler, 2014), which are not affected by this chemical treatment. Again, we would like to stress the fact that all marine particles (including foraminifera, radiolarians, diatoms, coccolithophores, etc.) have been removed from the samples prior to grain-size analysis. To verify if indeed all marine biogenic particles were removed, we performed microscope analysis (Figure 5). This confirmed that the fraction analyzed for particle size is the insoluble or lithogenic fraction, interpreted as mineral dust.

We modified the paragraph from the methods section addressing this issue.

*Page 6, lines 7-17:*

"The sediment traps collect all particles settling down into the ocean, that besides mineral dust includes the skeletons of marine plankton (foraminifera, radiolarians, diatoms, coccolithophores, etc.), organic matter (marine and aerosols from biomass burning) and potentially volcanogenic and cosmogenic particles (Plane, 2012). All these biogenic constituents were chemically removed in three steps to isolate the insoluble or lithogenic dust fraction from all samples, prior to grain-size analysis, following the procedure described by McGregor et al. (2009). Shortly, organic matter was oxidized using $H_2O_2$, followed by dissolving the biogenic carbonates using HCl, and removing biogenic silica by adding NaOH. What remains is the lithogenic fraction which is considered to consist mainly of dust, as confirmed by microscope analysis (Fig. 5). Indeed, some of the dust particles have a risk of being removed during this process including lithogenic carbonates and organic particles of non-marine origin. However, lithogenic carbonates are more resistant to the chemical treatment than the biogenic carbonates. Also, since this is the case for every sample analyzed, they can be compared directly to each other."

Other basic questions are still focused on the transport of dust from the surface downwards. Simple calculations of particle sedimentation rates (in the absence of any water currents) indicates that the short transport time between the 1200 m and 3500 m traps cannot be explained simply by particle setting. For example, for a 10 μm particle of density 2700 kg/m$^3$, a distance of only ~200 m would be traversed in a ~few weeks. Since 10 μm is close to the mass-modal diameter, it's clear that something else must be going on. Hence the interest in currents, I imagine, which the atmospheric community does not have any intuition about. Adding information about the rate and direction of currents will be very helpful. This is also relevant to the apparent discontinuity in results at

the lower traps and at the ocean floor; why would transport be very speedy from 1200 m to 3000m, but then essentially disappear in the lowest 1 - 2 km?

In the ocean, the dust particles deposited onto the ocean's surface do not settle individually, but as part of large marine particles like marine snow. These are aggregates > 500 µm of organic and inorganic particles of different composition and origin (e.g. Nowald et al., 2015). These particles have a much higher settling velocity, usually over 200 m day$^{-1}$, and the settling velocity may even increase with increased depth (Berelson, 2002;Nowald et al., 2015).

We modified the following lines in the revised manuscript:

*Page 5, lines 6-10:*

"These high settling velocities can be reached since the mineral dust particles are not deposited individually, but as part of large marine particles like marine snow. These are aggregates > 500 µm of organic and inorganic particles of different composition and origin (e.g. Nowald et al., 2015). These particles have settling velocities of approximately 200 m day$^{-1}$, and may increase with increased depth (Berelson, 2002;Nowald et al., 2015)."

In addition, current meters on the moorings show that, with few exceptions, current velocities remained well below < 12 cm s$^{-1}$, which is the threshold below which unbiased collection of settling particles occurs (Knauer and Asper, 1989). Above this threshold undertrapping occurs, with high current velocities resulting in a decreased particle flux in the sediment traps.

We've added the following lines to the revised manuscript:

*Page 3, lines 36-39:*

"Tilt-meters showed that the sediment traps remained nearly upright for the entire sampling period. With few exceptions, current velocities, as measured by current meters and ADCPs, remained well below 12 cm s$^{-1}$, the threshold below which unbiased collection of settling particles occur (Knauer and Asper, 1989)."

The apparent discontinuity in particle size between the sediment traps and the seafloor sediments is due to the difference in timing, and is not related to particle settling velocities: the sediment traps collect currently deposited dust over periods of 16 days, while the seafloor sediments contain dust deposited over 100s to 1000s of years, in a single sample. This is also extensively discussed in the manuscript.

Addressing these questions will help the atmospheric community contextualize the potential scale of any assumptions and uncertainties, and will help it grasp the significance of this manuscript as a step in achieving closure between atmospheric transport of mineral dust, and its removal from the atmosphere by dry and wet deposition.

As I had some concern that some portions of the reviewers' comments may not have been fully considered (these are the questions I reformed here; perhaps I am mistaken!), I have asked them, also, to comment on the response.

Thank you very much for working through this review process with ACPD; I have high expectations for the value of your technique and findings to contribute meaningfully to understanding and quantifying total and size-dependent dust transport and removal from the atmosphere.

PS - Minor additional comments:

It would be helpful to use clearer wording to describe the size distribution information throughout the paper. I found myself repeatedly questioning whether number or mass average values were being presented. For example, in the caption of Figure 3, the "average modal grain size" is presented. I think this is the average modal-mass grain size ? ; in Figure 4 the caption is for "average grain-size distributions" , but these represent average grain-volume distributions? Page 19 ,lines 3-7 : "...with modal particle diameters ranging..." could perhaps be more clearly stated "...with mass-modal particle diameters ranging..."? And so on.

The modal particle sizes described in the paper are neither number nor mass average values, but the modal values of the relative volume grain-size distributions. Average modal grain size means the average over several samples, in the case of Figure 3 grouped seasonally.

We modified the following lines in the revised manuscript:

*Page 8, lines 2-5:*

"Figure 3. Downwind fining and seasonality in average modal grain size per season for all seven traps (an average of all modal values from the relative volume grain-size distributions, grouped per season), for October 2012 – November 2013, and modal grain size of the seafloor sediments (from the grain-size distributions as show in Figure 4B), versus western longitude."

*Page 9, lines 3-4:*

"Figure 4. A: Average volume grain-size distributions of 24 samples from all seven sediment traps, where U = upper trap (1200 m) and L = lower trap (3500 m)."

On Figure 4, I'd also suggest a clarification of the vertical axis legend. As it looks like the width of the bins of the histogram are not one size in linear space, but one size in log-space, it is standard practice in the atmospheric community to specify the volume fraction as scaled by the log-space bin width (dV/dLog(D), the differential volume per horizontal step in log space). I imagine that this is what is shown already.

We have changed the title of the axis in Figures 4 and 6 (pages 9 and 11), and it now reads: "Frequency (vol. %) ; dV/dLog(D)".

Finally, my own elementary question reflecting lack of familiarity with your water-centric techniques: are the sampling volumes closed when not sampling/during recovery? Is there any chance that some vials sampled substantially longer times than others? If these are non-issues, that will be helpful to know.

The (24) sampling bottles of the sediment traps are mounted on a carrousel underneath the funnel of the trap. This carrousel has a zero-position, which is positioned underneath the funnel during deployment. The motor driving the carrousel is programmed to start sampling at a pre-defined date, such that all sediment traps along the whole transect start sampling simultaneously. All other bottles are sealed underneath the carrousel with O-rings. After the pre-programmed sampling interval, the carrousel switches one position and the next bottle is placed underneath the open funnel. After all bottles have sampled, the carrousel returns to the zero position until the sediment trap is recollected. The sediment-trap motor logs its actions, and from this data we can see that they performed flawlessly according to the pre-programmed intervals, for all seven sediment traps. This sediment-trap sampling method has been used in the marine community for several decades and proved efficient in many studies. Since no extraordinary events occurred, the precise mechanism of the sediment traps are not mentioned in the manuscript.

We modified the following lines in the revised manuscript:

*Page 3, lines 35-36:*

"All sediment traps operated synchronously over pre-programmed intervals of 16 days, and performed flawlessly."

**Co-Editor Decision: Reconsider after major revisions** (18 Aug 2016) by Joshua Schwarz
Comments to the Author:
Dear Dr. van der Does,

After having some time to consider the reviewers response to your revised manuscript, I have concluded that the required changes rise to the level of a major revision. In particular, both reviewers felt that the revised text should incorporate more of your responses. Indeed, I have always felt that manuscripts published in ACP should stand on their own without any need for readers to reference discussions in ACPD.

You can be very pleased that the reviewers also still agree on their assessment of the excellent promise of the scientific significance of your work. I very much look forward to another revision of your exciting manuscript! I

Regards,
Shuka
* * *
Dr. Joshua (Shuka) Schwarz
Guest Editor, ACP Special Edition: SALTRACE
NOAA ESRL CSD
Boulder, CO

The authors provided a revised version of the manuscript and point-to-point responses to the referees' comments. The new manuscript is indeed improved, and the authors answered satisfactorily to most of my previous comments. I think that some of those replies should also be incorporated more directly into the manuscript, to enhance the comprehension for all future readers. Here are my final (minor) comments.

3 / 9. "We used time-series FROM submarine sediment traps ..."
3 / 10. "of 16 days. Here we ..."

We modified the following lines in the revised manuscript:

*Page 3, lines 9-10:*

"We used time-series from submarine sediment traps moored at five locations along this transect, sampling synchronously over successive intervals of 16 days."

3 / 25-27. Not completely accurate (e.g. McGee et al., 2013)

The reviewer is correct that McGee et al. (2013) observed dust fluctuations at a higher frequency than "just" glacial-interglacial scales during the last 20kyr BP, however we wanted to sketch the bigger picture of the Late Quaternary. We now added the words "generally, throughout the Late Quaternary" to emphasize that we are really merely sketching the bigger picture throughout the whole Quaternary.

We modified the following lines in the revised manuscript:

*Page 3, lines 24-27:*

"In deep-sea sediments deposited offshore northwest Africa, Holz et al. (2004;2007), Mulitza et al. (2008) and Zühlsdorff et al. (2007) found links between dust deposition and variability in transport mechanisms, and generally, more dust deposition in dry glacial periods than in humid interglacials, throughout the Late Quaternary."

6 / 33-35. How are these two observations (one about a vertical profile, the other about a longitudinal gradient) connected?

We agree that this is an inconsistent sentence, with two sentences that are not linked.

We modified the following lines in the revised manuscript:

*Page 7, lines 17-23:*

"All traps show a "shoulder" towards the coarse end of the grain-size distribution, which is most prominent at station M5 (Fig. 4A). This shows that coarse particles are not only deposited at proximal locations, but also transported over great distances. These shoulders are also found in the seafloor sediments (Fig. 4B). They include "giant" particles of more than 100 µm, which are observed as far west as station M5 (57˚ W; approximately 4400 km from the African coast, and thus ever further from the actual dust source), and consist of both platy mica and rounded quartz particles (Fig. 5)."

9 / 10. I would suggest to explain already here the size distribution metrics that will be shown and discussed. For instance, explain that the modal diameter of particles-volume distributions was chosen to focus on the "main" dust mode, and that this particular metric does not account for the presence of shoulders. On the other hand you will show a complementary metric, i.e. sand%, to discuss the presence of shoulders.

We added a paragraph that discusses the used metrics at the beginning of the Results section.

*"The particle size of Saharan dust can be expressed in many ways. Showing grain-size distributions of all individual samples makes it more difficult to distinguish seasonal and spatial changes. Therefore, the modal particle size is chosen to represent the main dust mode, to better illustrate these changes. In addition, for station M1, the percentage of particles > 63 μm is shown, to highlight the coarse second peak, or "shoulder" of the grain-size distributions."*

13 / 12-15. This paragraph (together with the comment that deposition fluxes will be assessed separately) addresses my comment about giving at least an idea of the potential amounts of those large/giant particles. In their reply to my previous comment (8 / 11-15) the authors also seem to confirm that overall their counting statistics are high, since they have very concentrated samples (as opposed to ice cores). While in general I must agree with that, as expected, I still wonder if this is also the case for the tails in particular, which were the object of my concern. The microscope images indeed show the presence of those particles. I wish that the authors could provide some additional insight into the procedure to convince us that those are also a genuine signal from the water column (e.g. not contamination from the laboratory or similar). These data are are very promising, and the first of this kind, and many colleagues should be interested in those, therefore I think it's worth assessing potential problems in their interpretation, or more positively assuring their validity.

Indeed the particles represented in the tails or shoulders of the grain-size distributions are most likely only a few particles, which, due to their size, have a higher impact on the volume-size distribution than smaller particles. Still, when looking at the microscope images, a handful of these "giant" particles are observed in each sample, and when considering the samples are only 1/25 of the original sample, which sampled for 16 days and collected material over 1 m$^2$, we come to the conclusion that the total transport and deposition of these giant particles over the Atlantic Ocean must be substantial.

To illustrate these giant particles even better, and to show that they are present at every station along the transect, we've added more microscope images to Figure 5 in the revised manuscript (page 10). As with any analysis, contamination could happen at any stage, but this is of course limited as much as possible. All parts of the sediment trap are kept clean throughout the whole sampling procedure. The sampling bottles are cleaned thoroughly at the home institute (cleaned with 1N HCl and rinsed thoroughly with demineralized water) and then closed. Once onboard the ship, where the biocide is added, all handling of the bottles occurs in fume hoods so that we can be confident that contamination is minimal. Since these giant particles are present in almost every sample, at every station along the transect, the chance of these particles being contamination in every case is considered negligible.

We modified the following lines in the revised manuscript:

"However, we observed giant particles (≥100 μm) as far west as station M5 (57° W; approximately 4400 km from the African coast) (Fig. 5), and also mica particles, whose platy shape allows for aerial transportation over greater distances (Stuut et al., 2005). Only a handful of these coarse particles are found in the samples, however when considering these are 1/25 of the original samples, collecting sediments over 1 m$^2$ of ocean, over a time period of 16 days, this means that the amount of giant particles being transported over the Atlantic Ocean every year can be considered to be substantial. Such coarse particles are generally not incorporated into climate models (Kok, 2011).The underestimation of the coarse size fraction in climate models may have its origin in the sampling of dust of specific size classes, e.g. $PM_{10}$ and $PM_{2.5}$, which form the basis of the guidelines from the World Health Organization (WHO, 2006) on fine-grained particles."

14 / 10. I would make it more clear in the text that this is one possibility, as the authors point out in their replies. E.g. something like "Assuming a consistent relation in the vertical profiles (as discussed above), we speculate that this difference may be related to a change in conditions in the source areas."

We tried to highlight the fact that increased emission as a result of human activity due to commercial agriculture is one possible mechanism that could explain the difference in particle size between the sediment traps and seafloor sediments, by modifying the following lines in the revised manuscript:

*Page 14, lines 3-9:*

"Deposition of coarser dust is in line with increased emission as a result of human activity since the nineteenth century due to commercial agriculture (Mulitza et al., 2010). Not only does increased human activity in the source region increase dust emissions, it also enables larger particles to be emitted (McTainsh et al., 1997), which is one possible mechanism that could cause the particle size of the deposited Saharan dust to become gradually coarser over time, as we observe now in the sediment traps. However, we do not exclude other mechanisms that could be responsible for the change in particle size of mineral dust as deposited along the sampled transect."

19 / 7-9. This sentence is unclear

We modified the following sentence in the revised manuscript:

*Page 18, lines 26-29:*

"Coarser dust is found in the sediment traps as opposed to the seafloor sediments, which is in line with increased emission and coarser dust due to the onset of commercial agriculture in the 19$^{th}$ century, and is a possible explanation for the difference in particle size between the two records."

19 / 19. Didn't you suggest this is unlikely, based on HYSPLIT?

In this case, HYSPLIT does not show a distinct seasonal difference in air-layer trajectories, however this does not exclude changing source areas on a seasonal or (multi-)annual scale. HYSPLIT can serve as a good indicator of air-layer trajectories, however when calculated over longer time intervals these trajectories become more uncertain. In addition, from HYSPLIT it is not clear what surface conditions contribute to the uplift of particles into the long-distance transporting air layers. Still, differences in source areas could contribute to differences in composition and particle size of the dust.

We modified the following lines in the revised manuscript:

*Page 18, lines 9-13:*

"An increased number of coarse particles during spring could mean that the dust originates from a different source area. Backward trajectories calculated over the entire sampling period do not show this. However, these backward trajectories serve only as an indicator for air-layer trajectories, but from these it does not become clear what surface conditions contributed to the uplift of particles in the long-distance transporting air layers."

19 / 20. I would suggest "Multiple-year samples from this transect AND COUPLED DEPOSITION FLUX MEASUREMENTS should clarify"

We agree and modified the indicated sentence in the revised manuscript:

*Page 19, lines 1-3:*

"Multiple-year samples from this transect and coupled dust deposition fluxes should clarify which of the above mentioned processes are more dominant, in order to be incorporated into e.g. climate models and climate reconstructions."

References:
McGee, D., P. B. deMenocal, G. Winckler, J. B. W. Stuut, and L. I. Bradtmiller (2013), The magnitude, timing and abruptness of changes in North African dust deposition over the last 20,000 yr, Earth Planet. Sci. Lett., 371–372, 163–176, doi:10.1016/j.epsl.2013.03.054.

van der Does et al. present interesting results regarding deposition of dust across the Atlantic Ocean using moored traps. The potential for this study is significant, considering the cross disciplinary effort between sedimentology and atmospheric physics and transport. However, a dearth of information regarding the atmospheric component, even in this version, warrants further revision.

General comments:

I see that many of the authors' responses to the reviewer comments, although sufficient in providing the necessary explanations, did not result in any additions to the manuscript. As in this case, should the authors need to explain themselves to the reviewers for clarity, the readers will certainly need and appreciate the explanations as well. I suggest the authors take their responses from the first revision and incorporate them into the manuscript more than they currently have.

We have tried to incorporate more of the responses to the reviewer's comments in the manuscript. Please see the new revised manuscript with tracked-changes to see the most recent updates to the paper.

I continue to question the speculation that all the particles sized were dust based on the information provided. As an aerosol chemist, I observe quite a bit of spatial and vertical heterogeneity in atmospheric aerosols, even when close to a source region. Without chemical analysis, I find it difficult to believe all the particles from this study were dust, even considering the chemical treatment methods used to reduce the contribution from other aerosol types. Considering the work adds another variable to what could influence the aerosol composition, there could be more room for introducing complexity from additional particulate components from the ocean. With that said, there are a few things the authors could do to alleviate this issue as discussed below.

As we have tried to explain in the methods section of the paper, all the biogenic particles of marine origin are chemically removed: this includes all organic matter, (biogenic) carbonates and biogenic silica. What is left is the insoluble or lithogenic fraction, which is only sourced in atmospheric mineral dust. This may potentially include a small fraction of volcanogenic and cosmogenic particles as well (Plane, 2012), however on the other hand, carbonate dust particles and organic particles of non-marine origin are also removed from the samples by the chemical treatment applied. Since this is the case for every sample that we analyzed in this study, they can be compared directly to each other. In addition, we know that most of Saharan dust is made up of siliciclastics (e.g. Scheuvens and Kandler, 2014), which are not affected by this chemical treatment. Again, we would like to stress the fact that all marine particles (including foraminifera, radiolarians, diatoms, coccolithophores, etc.) have been removed from the samples prior to grain-size analysis. To verify if indeed all marine biogenic particles were removed, we performed microscope analysis (Figure 5). This confirmed that the fraction analyzed for particle size is the insoluble or lithogenic fraction, interpreted as mineral dust.

We modified the paragraph from the methods section addressing this issue.

*Page 6, lines 7-17:*

"The sediment traps collect all particles settling down into the ocean, that besides mineral dust includes the skeletons of marine plankton (foraminifera, radiolarians, diatoms, coccolithophores, etc.), organic matter (marine and aerosols from biomass burning) and potentially volcanogenic and cosmogenic particles (Plane, 2012). All these biogenic constituents were chemically removed in three steps to isolate the insoluble or lithogenic dust fraction from all samples, prior to grain-size analysis, following the procedure described by McGregor et al. (2009). Shortly, organic matter was oxidized using $H_2O_2$, followed by dissolving the biogenic carbonates using HCl, and removing biogenic silica by adding NaOH. What remains is the lithogenic fraction which is considered to consist mainly of dust, as confirmed by microscope analysis (Fig. 5). Indeed, some of the dust particles have a risk of being removed during this process including lithogenic carbonates and organic particles of non-marine origin. However, lithogenic carbonates are more resistant to the chemical treatment than the biogenic carbonates. Also, since this is the case for every sample analyzed, they can be compared directly to each other."

There are ways to support conclusions regarding aerosol type, such as the authors do with MODIS. However, MODIS also has its limitations. For instance, in Figure 10, AOD is clearly enhanced in a pathway from Africa, but

these aerosols could be dust or emissions from biomass burning, which is especially predominant in the summer and is evident by the high AOD propagating off central Africa. Did the authors account for biomass burning aerosol? Have the authors considered adding CALIPSO to their analysis to evaluate aerosol type, at least between the coast of Africa and M1? This would greatly reduce the assumptions made regarding aerosol type.

Indeed, some mixing of aerosols (mineral dust and aerosols from biomass burning) could occur (Adams et al., 2012), and it could be possible that these aerosols resulting from biomass burning are collected within the sediment traps. However, since they are of organic origin, they are chemically removed prior to grain-size analysis, using Hydrogen Peroxide (see also the Methods section, page 6 of the revised manuscript), so these are not accounted for in the grain-size distributions as presented in this paper. Based on CALIPSO data, Adams et al. (2012) speculate that smoke is not transported over great distances, opposed to mineral dust. Figure 10 is mainly used to illustrate the seasonal latitudinal movement of the dust cloud. It shows that AOD values are highest during summer, not only over the studied transect but also more to the south, between the equator and 10 °S, the latter possibly being related to biomass burning. We do not make a link between AOD and dust deposition fluxes, since these data are not available for this paper.

We added the following lines to the revised manuscript:

Page 6, lines 7-12:

"The sediment traps collect all particles settling down into the ocean, that besides mineral dust includes the skeletons of marine plankton (foraminifera, radiolarians, diatoms, coccolithophores, etc.), organic matter (marine and aerosols from biomass burning) and potentially volcanogenic and cosmogenic particles (Plane, 2012). All these biogenic constituents were chemically removed in three steps to isolate the insoluble or lithogenic dust fraction from all samples prior to grain-size analysis, following the procedure described by McGregor et al. (2009)."

*Page 14, lines 15-17:*

"However, the aerosols over our study area are mostly mineral dust originating from the African continent (Yu et al., 2015), also since Adams et al. (2012) speculate that smoke is not transported over great distances, as opposed to mineral dust."

HYSPLIT is a great tool, but when used cautiously. Typically, the error in a single trajectory results in 30% of its distance, meaning the error in where the air mass traveled can be quite substantial far from its initiation location. This can be alleviated by using an ensemble of HYSPLIT trajectories. However, only 4 trajectories are shown, which are also used to conclude in the abstract that transport occurs at higher altitudes and fast winds. Evidence supporting this conclusion is not provided, thus making it quite speculative. The authors do state in the responses that they did this for the year at M1 and it resulted in a mess of sources, however, these trajectories can be clustered. Showing clusters of trajectories, perhaps at various seasons and locations would increase the statistical significance and reduce the uncertainty in the sources and transport pathways, and better support their arguments.

The HYSPLIT trajectories shown in the paper are only a means to illustrate a typical winter and summer dust transport scenario. These differences in dust transportation altitude between the different seasons have been demonstrated in literature before (e.g. Tsamalis et al., 2013;Adams et al., 2012), and this was used to interpret the data of the present paper. Figure 9A and -B also illustrate the different transport altitudes between summer and winter. In the caption of Figure 9 we also note that these are "typical examples" of summer and winter dust transport, as described in the cited literature.

It is true that sea salt is soluble, but mineral surfaces have been shown to become coated by these soluble species in solution (REFS) and at times can be an irreversible reaction, as been shown in cloud droplets (CZICZO). Considering these particles are much larger than aerosols studied in clouds, there is substantial surface area for these (sometimes) irreversible reactions to take place. How are the authors so sure that the treated dust particles are devoid of soluble components, or other types of particles?

Without a comprehensive chemical analysis we have not verified this, however the microscope images show giant quartz particles with no apparent coatings. For further demonstration of the giant particles in the samples, we have added more microscope images to Figure 5 (page 10 of the revised manuscript). This also illustrates that these giant particles are deposited at every station along the transect, however possibly decreasing in amount towards the west.

Overall, providing more sound, empirical evidence in addition to more statistics on air mass transport would support the authors' conclusion that what they are sizing is most likely dust. At the very least the authors should clearly highlight the assumption that these particles are dust throughout the manuscript, and provide some more background on what other particulate components could be contributing to the samples. Providing examples of previous studies evidencing dust as the major component in this region would also help.

Since these samples were obtained by submarine sediment traps, the bulk samples exist of mainly marine particles, including organic matter, planktonic biomineral shells (e.g. foraminifera, radiolarians, diatoms, coccolithophores, etc.) next to mineral dust, and potentially volcanogenic and cosmogenic particles (Plane, 2012). After the chemical treatment, in the form of sequential leaching, all biogenic particles are removed from the sample and what is left is solely the insoluble, lithogenic fraction, which is almost entirely made up of mineral dust (see also the reply to the first question). As Yu et al. (2015) have pointed out in their study, 182 million tons of Saharan dust are transported from Africa over the Atlantic Ocean, which is calculated to amount to 5 g/m$^2$/yr. This considerable amount evidences dust as the major lithogenic component in the Atlantic Ocean. As mentioned also in the first rebuttal, lithogenic input other than mineral dust is believed to be negligible.

Were any of the extracted dust samples weighed prior to and following chemical treatment to remove non-dust components? This would strengthen the argument that what was left was likely dust, i.e., if total mass concentrations decreased after treatment. Would also provide a more quantitative approach as compared to Figure 2.

As mentioned before, dust mass fluxes would be a valuable addition to the data presented in this paper, but this will be the scope of another paper. However, some masses have been determined. As an example, for station M1, the average weight of the total bulk sample (so including all marine particles) is close to 1800 mg. The average weight of the isolated lithogenic (dust) fraction of the samples is nearly 30 mg. This means that the lithogenic fraction is about 1.7 % of the total sample, and proving the leaching procedure very effective. Figure 2 is intended to illustrate the similarity between the upper (1200 m) and lower (3500 m) traps, and demonstrate the settling velocity of the sediments.

The authors do not provide any discussion of or show uncertainty in the measurements in the manuscript. Please show error bars in the figures and clearly discuss the possible sources of uncertainty or variability in the measurements. For example, Figure 3 shows several markers close together, but it is unknown if these disparities are significant or simply within error. It would be more difficult to draw the conclusions discussed by the authors should these all be within error of each other.

For this study, we did not analyze any duplicate samples, since there simply are not more samples to be analyzed. The reproducibility is checked regularly at MARUM, Bremen (Germany), where author JBS holds a part-time position, using exactly the same equipment set-up as in this study, including the same laser particle sizer and degassed-water system, by Dr. J. Titschack. This is done by replicate analyses of three internal glass-bead standards, and the reproducibility is found to be better than ± 0.7 µm for the mean and ± 0.6 µm for the median particle size (1σ). The average standard deviation integrated over all the size classes is better than ± 4 vol %.

The markers in Figure 3 are an average of seasons (3 months), covering in total 5 to 7 samples per season. The more accurate difference between the seasons can be observed when looking at Figure 7 (Figures 7 – 9 of previous version), which shows more accurately that there is indeed a significant difference between the different seasons. Figure 3 is also meant to illustrate the difference between the sediment trap samples and the seafloor sediment samples, the latter being finer grained.

We have added some lines to the Methods section about the reproducibility of the laser particle sizer.

"The reproducibility is checked regularly at MARUM, Bremen (Germany), using exactly the same equipment set-up as in this study, by Dr. J. Titschack. This is performed by replicate analyses of three internal glass-bead standards, and the reproducibility is found to be better than $\pm$ 0.7 µm for the mean and $\pm$ 0.6 µm for the median particle size (1$\sigma$). The average standard deviation integrated over all the size classes is better than $\pm$ 4 vol %."

There is quite a bit of redundancy in some of the figures (namely 7, 8, and 9). The authors should consider combining these, which will serve to eliminate the redundancy and a more direct comparison.

We initially chose to separate these figures to avoid a figure becoming too crowded. Plotting modal grain sizes of all traps (N=7) together in one plot will result in an unclear image. Separating the upper and lower traps made it harder to compare these for stations M2 and M4, where both are present, thus resulting in Figure 9. However, we have chosen now to combine Figures 7, 8, 9 and 12, as Figure 7A and B in the revised manuscript (page 12).

Specific comments:

Page 5, lines 11-12: Elaborate on this. Where was chlorophyll and salinity higher? This should perhaps be shown in a supporting information file, especially considering these can change greatly on a seasonal basis throughout the year.

Data of Chlorophyll and salinity can track fresh-water input and productivity from major rivers in the ocean. As seen from this data, the influences at the sampling stations seem to be minor. These graphs (see below) are added as Supplement to the manuscript.

We modified the following lines to the revised manuscript:

Page 5, lines 19-20:

"Limited influence of major rivers is also visible when looking at (satellite) data of chlorophyll or salinity (see Supplement)."

[Figure]

[Figure]

Adams, A. M., Prospero, J. M., and Zhang, C.: CALIPSO-Derived Three-Dimensional Structure of Aerosol over the Atlantic Basin and Adjacent Continents, Journal of Climate, 25, 6862-6879, 10.1175/jcli-d-11-00672.1, 2012.

Berelson, W. M.: Particle settling rates increase with depth in the ocean, Deep-Sea Research Part Ii-Topical Studies in Oceanography, 49, 237-251, 2002.

Knauer, G., and Asper, V.: Sediment Trap Technology and Sampling: U.S. Global Ocean Flux Study, WHOI, U.S. GOFS Planning Report, 1989.

McGee, D., deMenocal, P. B., Winckler, G., Stuut, J. B. W., and Bradtmiller, L. I.: The magnitude, timing and abruptness of changes in North African dust deposition over the last 20,000 yr, Earth and Planetary Science Letters, 371, 163-176, 10.1016/j.epsl.2013.03.054, 2013.

[revised manuscript text omitted]

Here we provide maps of Chlorophyll A concentrations and Sea Surface Salinity, over the sampled period October 2012 – November 2013, over the equatorial Atlantic Ocean.